# Activity regulates a cell type-specific mitochondrial phenotype in zebrafish lateral line hair cells

Andrea McQuate[1,2]*, Sharmon Knecht[1], David W Raible[1,2]*

[1]Department of Biological Structure, University of Washington, Seattle, United States; [2]Department of Otolaryngology-HNS, University of Washington, Seattle, United States

**Abstract** Hair cells of the inner ear are particularly sensitive to changes in mitochondria, the subcellular organelles necessary for energy production in all eukaryotic cells. There are over 30 mitochondrial deafness genes, and mitochondria are implicated in hair cell death following noise exposure, aminoglycoside antibiotic exposure, as well as in age-related hearing loss. However, little is known about the basic aspects of hair cell mitochondrial biology. Using hair cells from the zebrafish lateral line as a model and serial block-face scanning electron microscopy, we have quantifiably characterized a unique hair cell mitochondrial phenotype that includes (1) a high mitochondrial volume and (2) specific mitochondrial architecture: multiple small mitochondria apically, and a reticular mitochondrial network basally. This phenotype develops gradually over the lifetime of the hair cell. Disrupting this mitochondrial phenotype with a mutation in *opa1* impacts mitochondrial health and function. While hair cell activity is not required for the high mitochondrial volume, it shapes the mitochondrial architecture, with mechanotransduction necessary for all patterning, and synaptic transmission necessary for the development of mitochondrial networks. These results demonstrate the high degree to which hair cells regulate their mitochondria for optimal physiology and provide new insights into mitochondrial deafness.

*For correspondence:
amcquate@uw.edu (AMcQ);
draible@uw.edu (DWR)

**Competing interest:** The authors declare that no competing interests exist.

## Editor's evaluation

This valuable study of serial block-face scanning electron microscopy on zebrafish lateral line hair cells provided compelling cellular evidence for the importance of normal hair cell function in establishing mitochondrial patterning. This work will be of broad interest to cell biologists studying mitochondrial function.

## Introduction

Mitochondria are essential subcellular organelles in nearly all eukaryotic cells, where they perform and regulate manifold functions, including ATP production, calcium buffering, apoptosis, metabolite generation, among others. These functions are influenced by a cell's total mitochondrial volume, regulated by mitochondrial biogenesis (*Jornayvaz and Shulman, 2010*) and subsequent mitochondrial architecture, sculpted by mitochondrial fusion and fission (*Picard et al., 2013*). Mitochondrial fusion and elongated mitochondria are associated with heightened mitochondrial membrane potentials, increased ATP production, and improved calcium buffering (*Picard et al., 2013*; *Szabadkai et al., 2006*; *Gomes et al., 2011*). Meanwhile, mitochondrial fission and smaller mitochondria are associated with lower mitochondrial membrane potentials, lower ATP production, and apoptosis (*Liu et al., 2020*). The combination of these features produces an overall mitochondrial phenotype according

**eLife digest** Our ability to perceive sounds relies on tiny cells deep inside our ears which can convert vibrations into the electrical signals that our brain is able to decode. These 'hair cells' sport a small tuft of short fibers on one of their ends that can move in response to pressure waves. The large amount of energy required for this activity is provided by the cells' mitochondria, the small internal compartments that act as cellular powerhouses. In fact, reducing mitochondrial function in hair cells can lead to hearing disorders.

Mitochondria are often depicted as being bean-like, but they can actually adopt different shapes based on the level of energy they need to produce. Despite this link between morphology and function, little is known about what mitochondria look like in hair cells. Filling this knowledge gap is necessary to understand how these structures support hair cells and healthy hearing.

To address this question, McQuate et al. turned to zebrafish, as these animals detect vibrations in water through easily accessible hair cells on their skin that work just like the ones in the mammalian ear. Obtaining and analysing series of 3D images from a high-resolution microscope revealed that hair cells are more densely populated with mitochondria than other cell types. Mitochondrial organisation was also strikingly different. The side of the cell that carries the hair-like structures featured many small mitochondria; however, on the opposite side, which is in contact with neurons, the mitochondria formed a single large network. The co-existence of different types of mitochondria within one cell is a novel concept.

Further experiments investigated how these mitochondrial characteristics were connected to hair cell activity. They showed that this organisation was established gradually as the cells aged, with cellular activity shaping the architecture (but not the total volume) of the mitochondria.

Overall, the work by McQuate et al. provides important information necessary to develop therapeutics for hearing disorders linked to mitochondrial dysfunction. However, by showing that various kind of mitochondria can be present within one cell, it should also inform studies beyond those that focus on hearing.

to cellular need. Failure to achieve an appropriate mitochondrial phenotype results in a variety of pathologies, particularly in highly metabolically active cells such as those that are electrically excitable (*Reddy et al., 2011*).

Hair cells (HCs) in the peripheral auditory nervous system mediate hearing and balance. Deflection of the stereocilia bundle at the apical pole of the HC results in cation influx in a process known as mechanotransduction, and subsequent depolarization and calcium influx through voltage-gated calcium channels (cav1.3) results in glutamate release from ribbon synapses at the basolateral pole onto afferent neurons. HCs heavily depend on mitochondria to sustain energetic demands, with 75% of their ATP usage produced via oxidative phosphorylation (*Puschner and Schacht, 1997*). It is perhaps due to their high dependency on mitochondria that HCs are particularly susceptible to mitochondrial alterations; mitochondria are implicated in both hereditary and environmentally induced hearing loss, as well as aging (*Kokotas et al., 2007*; *Someya and Prolla, 2010*; *Böttger and Schacht, 2013*). Mutations in over 30 mitochondrial-associated genes result in hearing loss in humans. These include mutations in the gene *opa1*, necessary for mitochondrial fusion (*Leruez et al., 2013*; *Liguori et al., 2008*).

Mitochondria are implicated at both poles of healthy HCs. In rat cochlear inner HCs, apical mitochondria have been shown to buffer calcium influx during mechanotransduction (*Beurg et al., 2010*; *Pickett et al., 2018*). Meanwhile, in zebrafish lateral line HCs, basal mitochondrial calcium uptake is essential for regulating ribbon size (*Wong et al., 2019*). Mitochondria also play a role in HC vulnerability to aminoglycoside exposure. HCs that have been treated with neomycin demonstrate abnormal mitochondrial morphologies prior to other insults (*Owens et al., 2007*). Neomycin-induced HC death requires mitochondrial calcium uptake (*Esterberg et al., 2014* and *Esterberg et al., 2016*), and HC sensitivity to neomycin increases with cumulative mitochondrial activity (*Pickett et al., 2018*). Similarly, calcium import into the mitochondria via the MCU has been implicated in noise-induced hearing loss (*Wang et al., 2018*), and related mitochondrial potentials are disrupted in aging (*Perkins et al., 2020*).

**Table 1.** Individual cells and mitochondria reconstructed.

| Genotype | Age (dpf) | Fish | Neuromasts | Cells | Mitochondria |
|----------|-----------|------|------------|-------|--------------|
| WT HCs | 3 | 2 | 2 | 12 | 299 |
| WT HCs | 5–6 | 3 | 5 | 65 | 2347 |
| *cdh23* HCs | 5 | 2 | 4 | 19 | 382 |
| *cav1.3* HCs | 5 | 2 | 4 | 48 | 1939 |
| *opa1* HCs | 5 | 1 | 1 | 5 | 778 |
| | | | | | |
| WT SCs | 5 | 2 | 3 | 13 | 163 |
| Total | | | | 162 | 5908 |

dpf: days post fertilization; HC: hair cell.

Given the known impact of mitochondrial structure on their function, understanding the detailed morphological characteristics of HC mitochondria is a necessary first step for interpreting how these morphologies intersect with HC physiology and vulnerability (*Lesus et al., 2019*). We hypothesized that HCs maintained their mitochondria in an optimal configuration dependent on mitochondrial fusion to sustain high metabolic demands. Here, we detailed the characteristics of mitochondria in the HCs of the zebrafish lateral line. The lateral line is composed of clusters of HCs and surrounding supporting cells (SCs) called neuromasts (NMs) found on the surface of the fish's body and detects changes in water flow. This information is vital for schooling and feeding behaviors. Lateral line HCs are genetically and morphologically similar to the HCs located within the mammalian inner ear, but are more easily accessible to genetic manipulations and experimentations (*Pickett and Raible, 2019*; *Sheets et al., 2021*). As fluorescent markers lack the resolution to distinguish between individual organelles, we turned to serial block-face scanning electron microscopy (SBFSEM) to produce three-dimensional reconstructions of HCs and their mitochondria with ultrastructural resolution. Over the course of this study, we reconstructed 5908 individual mitochondria from 162 different cells (16 NMs, 10 fish, *Table 1*). We demonstrated that zebrafish lateral line HCs had a mitochondrial phenotype distinct from SCs. This phenotype included a high total mitochondrial volume, and a particular architecture, with large, highly networked mitochondria at the basolateral pole near the synaptic ribbons, and smaller mitochondria positioned apically. This phenotype developed with HC maturation and was dependent on mechanotransduction and synaptic activity. Overall, our results demonstrate that HCs developed a highly specialized mitochondrial architecture sculpted by cell activity, which may explain their sensitivity to mitochondrial perturbation. Furthermore, the SBF datasets for each of the NMs used in this study have been made openly available, providing a comprehensive resource for further studying lateral line HCs and their organelles at the ultrastructural level.

## Results
### Hair cells contain a dense mitochondrial population not found in supporting cells

Although the importance of mitochondria in HCs is well-established, it is unclear how HC mitochondria compare with the mitochondria of other cell types, both in number and morphology. We used SBFSEM to reconstruct HCs and SCs from zebrafish anterior lateral line NMs (*Figure 1A*). This technique provides sufficient resolution to distinguish between individual mitochondria and compare individual mitochondrial morphologies. HC bodies and HC mitochondria were reconstructed via manual segmentation at 5–6 dpf, an age when the lateral line has completed maturation (*Figure 1B*, 3 fish, 5 NMs, total of 65 HCs). HCs were distinguishable from other cell types in the NM by the presence of synaptic ribbons, stereocilia, and kinocilia. To compare HC mitochondrial values to those of other cell types, we reconstructed the mitochondria of both peripheral and central SCs (three NMs from two fish, six central and seven peripheral SCs). SCs are a vital component of NMs, both structurally and physiologically. We reconstructed central and peripheral SCs, which have been found to have differing

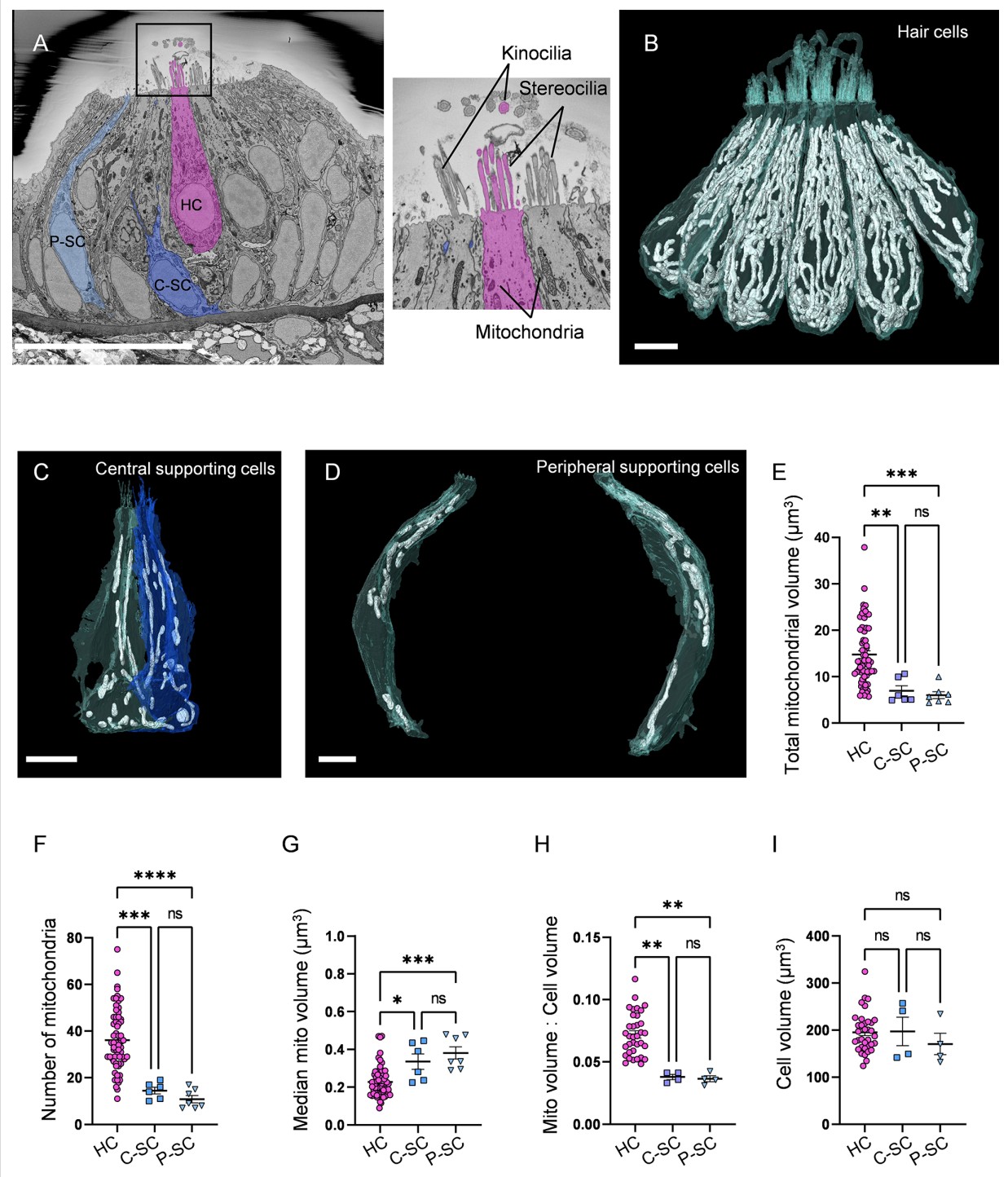

**Figure 1.** Hair cells (HCs) contain a higher mitochondrial volume than supporting cells. (**A**) SEM cross-section through 5 days post fertilization (dpf) zebrafish neuromasts (NM) (NM3, *Figure 1—source data 2*). Scale bar = 20 μm. Inset shows the stereocilia bundle and kinocilium labeled for 1 HC. (**B**) Six reconstructed HCs from NM3, with mitochondria shown in white. Scale bar = 5 μm. (**C**) Two central supporting cells (C-SCs) reconstructed from NM3. Scale bar = 5 μm. (**D**) Two peripheral supporting cells (P-SCs) reconstructed from NM3. Scale bar = 3.5 μm. (**E**) Sum of mitochondrial volume for HCs, C-SCs, and P-SCs. (In μm$^3$) HC: 14.8 ± 0.8; C-SC; 6.9 ± 1.1, P-SC; 6.0 ± 0.7. (**F**) Number of individual mitochondria in HCs, C-SCs, and P-SCs. HC: 36.1 ± 1.6; C-SC: 14.5 ± 1.4; P-SC: 10.9 ± 1.5. (**G**) The median mitochondrial volume in HCs, C-SCs, and P-SCs. HC: 0.2 ± 0.01; C-SC: 0.3 ± 0.04; P-SC: 0.4 ± 0.03. (**H**) The ratio of the total mitochondrial volume to the total cell volume in HCs, C-SCs, and P-SCs. HC: 0.07 ± 0.003; C-SC: 0.04 ± 0.002; P-SC: 0.04 ± 0.002. (**I**) The cell volume of HCs, C-SCs, and P-SCs. HC: 195.6 ± 7.2; C-SC: 197.4 ± 30.05; P-SC: 170.4 ± 22.7. Kruskal–Wallis test with Dunn's multiple

*Figure 1 continued on next page*

*Figure 1 continued*

comparisons, *p<0.05, **p<0.01, ***p<0.001, ****p<0.0001. For (**E–G**), HC: n = 65, 5 NMs, 3 fish; C-SC: n = 6, 3 NMs, 2 fish; P-SC: n = 7, 3 NMs, 2 fish. For (**H, I**), HCs: n = 35, 3 NMs, 3 fish; C-SC: n = 4, 2 NMs, 2 fish; P-SC: n = 4, 2 NMs, 2 fish. Data are presented as the mean ± SEM.

The online version of this article includes the following source data for figure 1:

**Source data 1.** Raw values used in *Figure 1*.

**Source data 2.** Datasets used in *Figure 1*.

roles within the NM; peripheral SCs symmetrically divide to become HC progenitors during homeostasis and regeneration. Meanwhile, central SCs serve a 'glial-like' function in maintaining ion balance (*Thomas and Raible, 2019*; *Romero-Carvajal et al., 2015*). Central SCs (*Figure 1C*) were defined as interdigitating between two HCs, while peripheral SCs (*Figure 1D*) touched one or no HCs.

HCs contained on average a total mitochondrial volume of 14.8 ± 0.8 µm³ distributed across 36.1 ± 1.6 individual mitochondria (*Figure 1E and F*). The HC median mitochondrion volume was 0.2 ± 0.01 µm³ (*Figure 1G*). The average ratio of HC mitochondrial volume to cell volume was approximately 7% (*Figure 1H*). Both types of SCs had less total mitochondrial volume than HCs. We found central SCs contained a total mitochondrial volume of 6.9 ± 1.1 µm³ distributed over 14.5 ± 1.4 individual mitochondria (*Figure 1E and F*), with a median mitochondrial volume of 0.3 ± 0.04 µm³ (*Figure 1G*). Similarly, peripheral SCs contained a total mitochondrial volume of 6.0 ± 0.7 µm³ distributed over 10.9 ± 1.6 individual mitochondria with a median mitochondrial volume of 0.4 ± 0.03 µm³ (*Figure 1E–G*). In both central and peripheral SCs, the ratio of mitochondrial volume to cell volume averaged around 4% (*Figure 1H*). By contrast, overall cell volumes of HCs and both SC types were not different (*Figure 1I*). These data demonstrate that HCs have elevated mitochondrial volume and number relative to SCs.

## Mitochondrial architecture develops with hair cell maturation

We next asked how HC mitochondria change during cellular maturation. In the zebrafish lateral line, HCs undergo homeostatic turnover within NMs, where older, dying HCs are replaced when HC progenitors symmetrically divide to produce two new daughter HCs. As a result, NMs contain a spectrum of HCs of different ages. We used the length of the tallest stereocilium (stereocilia length) and length of the kinocilium to approximate the age of each individual HC as these both grow longer as HCs mature (*Kindt et al., 2012*). The actin-based stereocilia bundle contains HC tip links and mechanotransductive channels. The kinocilium is a microtubule-based structure that in the zebrafish lateral line has a role in HC development and establishing mechanotransduction. While some mammalian HCs shed their kinocilium, in zebrafish HCs the kinocilium continues to grow throughout the cell's lifespan. The lengths of both the stereocilia and kinocilium confirmed that the HCs we analyzed span the range of development (*Figure 2—figure supplement 1A and B*). The height of the stereocilia bundle and length of kinocilium also demonstrated a significant, positive correlation (*Figure 2—figure supplement 1C*). These metrics can then be used to relate HC age and HC mitochondrial properties.

As HCs mature, their mitochondria increase in volume and complexity (*Figure 2A and A'*). We found a positive correlation between HC stereocilia length and total mitochondrial volume (*Figure 2B*, p<0.0001) and the number of individual mitochondria (*Figure 2C*, p=0.003). Similar trends were found when kinocilium length was used to approximate HC age (*Figure 2—figure supplement 1D and E*). Because complete stereocilia bundles were more readily preserved than complete kinocilia over SBF serial sectioning, we focused on stereocilia length for the remainder of this study. These observations demonstrate that mitochondrial number and volume continue to grow as HCs mature.

Next, we examined the uniformity of the HC mitochondrial population. We found that over the course of maturation, HCs often developed a single, large, interconnected mitochondrion (max mito, *Figure 2A'*) that steadily expanded in volume (*Figure 2E*). In contrast, the volume of the median mitochondrion stayed relatively constant over HC development (*Figure 2D*). Therefore, the number of standard deviations between the max mito volume and the mean mitochondrial volume (max mito z-score) steadily increased during HC maturation (*Figure 2F*). By comparison, the mitochondrial populations of both central and peripheral SCs were more homogenous (*Figure 2G*). The distribution of individual mitochondrion volumes for representative mature HCs is shown in *Figure 2—figure supplement 2*. As HCs mature, the max mito localized to the basolateral pole of the HC, such that on average 53% of its total volume was localized in the bottom-most quadrant of the HC (*Figure 2H–J*). These

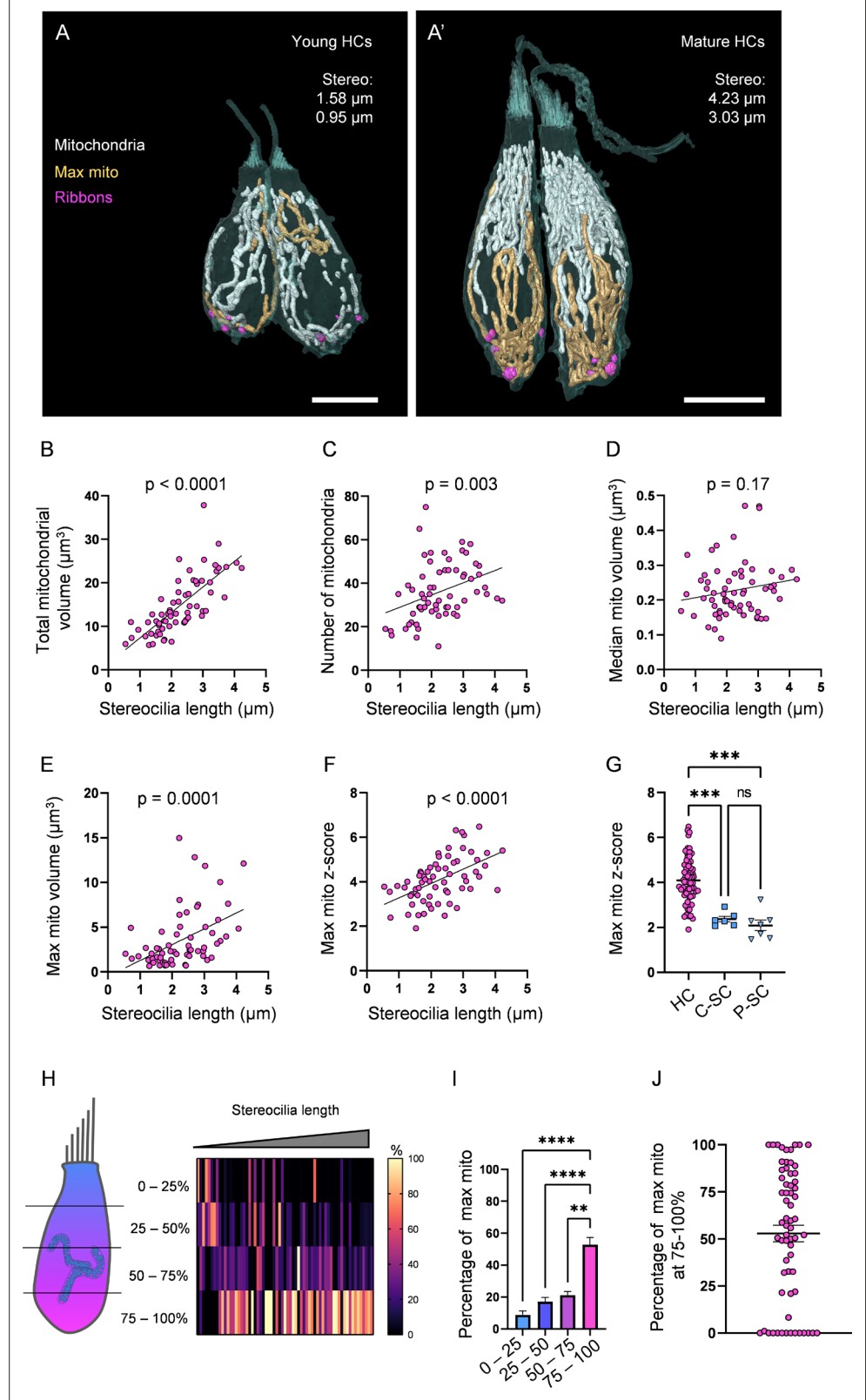

**Figure 2.** Hair cells (HCs) gradually develop a high mitochondrial volume with specific architecture. (**A**) Two young HCs from a 5 days post fertilization (dpf) neuromasts (NM) (NM1, **Figure 2—source data 2**). Mitochondria are shown in white. Single largest mitochondrion (max mito) is shown in gold. Synaptic ribbons shown in purple. Scale bar = 5 μm. (**A'**) Two mature HCs from a 5 dpf NM (NM3, different HCs than in **Figure 1B**). Scale bar = 6 μm.

*Figure 2 continued on next page*

*Figure 2 continued*

(**B**) Relationship between HC stereocilia length and the total mitochondrial volume. (**C**) Relationship between HC stereocilia length and the number of individual mitochondria. (**D**) Relationship between HC stereocilia length and the volume of the median mitochondrion. (**E**) Relationship between HC stereocilia length and the volume of the largest mitochondrion (max mito). (**F**) Relationship between HC stereocilia length and the number of standard deviations between the max mito and average mitochondrial volume (max mito z-score). Lines represent standard linear regression, with significance as indicated. (**G**) The z-score of the max mito in HCs, central supporting cells (C-SCs), and peripheral supporting cells (P-SCs) (mean ± SEM) HC: 4.1 ± 0.1; C-SC: 2.4 ± 0.1, P-SC: 2.1 ± 0.2. Kruskal–Wallis test with Dunn's multiple comparisons, ***p<0.001. (**H**) The percentage of the max mito located within each quadrant of an HC represented as a heat map. The length of each HC was normalized and broken into quadrants, with the highest HC point the base of the stereocilia bundle and the lowest point the lowest ribbon. The number of max mito segmentation coordinates within each quadrant of an HC were counted and represented as a percentage of all max mito coordinates. Cells are presented in order of their stereocilia lengths. (**I**) Summary of heat map data shown in (**H**). Most apical quadrant (0–25%): 9 ± 2.4%; 25–50%: 17.1 ± 2.7%; 50–75%: 21 ± 2.2%; Most basal quadrant (75–100%): 52.8 ± 4.4%. Kruskal–Wallis test with Dunn's multiple comparisons, **p<0.01, ****p<0.0001. (**J**) Percentage of the max mito located within the most basal quadrant for individual HCs. HC: n = 65, 5 NMs, 3 fish; C-SC: n = 6, 3 NMs, 2 fish; P-SC: n = 7, 3 NMs, 2 fish.

The online version of this article includes the following source data and figure supplement(s) for figure 2:

**Source data 1.** Raw values used in *Figure 2*.

**Source data 2.** Datasets used in *Figure 2*.

**Figure supplement 1.** Hair cell (HC) kinocilia and stereocilia lengths as representatives of cell age.

**Figure supplement 1—source data 1.** Raw values used in *Figure 2—figure supplement 1*.

**Figure supplement 2.** Distribution of individual mitochondrion volumes in mature hair cells (HCs).

**Figure supplement 2—source data 1.** Raw values used in *Figure 2—figure supplement 2*.

data show that as HCs develop, the mitochondrial population becomes nonuniform, with smaller mitochondria positioned apically, and a max mito localized to the base of the HC.

## Ribbon growth parallels mitochondrial growth and localization

HC mitochondria are known to regulate ribbon volume, and their ability to buffer calcium (*Wong et al., 2019*) and to generate ATP (*Stowers et al., 2002*; *Perkins et al., 2010*) at the basal end of HCs suggest they have roles in synaptic transmission. We therefore asked whether there was a relationship between HC mitochondrial development and ribbon development. We first measured ribbon volume and number across HC development. Averaging across all ages, HCs contained 5.5 ± 0.2 ribbons, with an average ribbon volume of 0.1 ± 0.006 μm$^3$, for a total ribbon volume of 0.6 ± 0.03 μm$^3$ (*Figure 3—figure supplement 1A–C*). The total ribbon volume steadily expanded over HC maturation (*Figure 3A and B*). The increase in total ribbon volume was primarily attributed to an increase in individual ribbon volume (*Figure 3C*) as there was no significant change in ribbon number over maturation (*Figure 3D*). We next compared ribbon development to mitochondrial development. We found a strong, positive correlation between the total mitochondrial volume and the total ribbon volume of each HC (*Figure 3E*). Given the localization of the max mito to the basolateral pole, we then asked whether the max mito was specifically associated with synaptic ribbons by calculating the average minimum geometric distance between the max mito and each ribbon per HC. We found that during HC maturation, the max mito localized to the synaptic ribbons, as reflected in the nonlinear decrease in the average minimum distance (*Figure 3F*). Meanwhile, there was no change in the position of the median mitochondrion relative to the ribbons during HC maturation (*Figure 3G*). Averaging across all HCs, the max mito was consistently closer to the synaptic ribbons (average, 1.9 ± 0.3 μm) than the median mitochondrion (6.3 ± 0.3 μm) (*Figure 3H*). These data show that during HC maturation, the size of individual ribbons increases in tandem with mitochondrial volume, and that the max mito becomes increasingly associated with ribbon synapses.

## Maturity of the neuromast affects mitochondrial architecture

To further explore patterns of mitochondrial maturation with HC development, we next examined HCs in immature 3 dpf NMs (2 NMs, 2 fish, total 12 HCs, *Figure 4A*), which, while shown to be functional, primarily demonstrate young biophysical profiles (*Olt et al., 2014*). Consistent with this

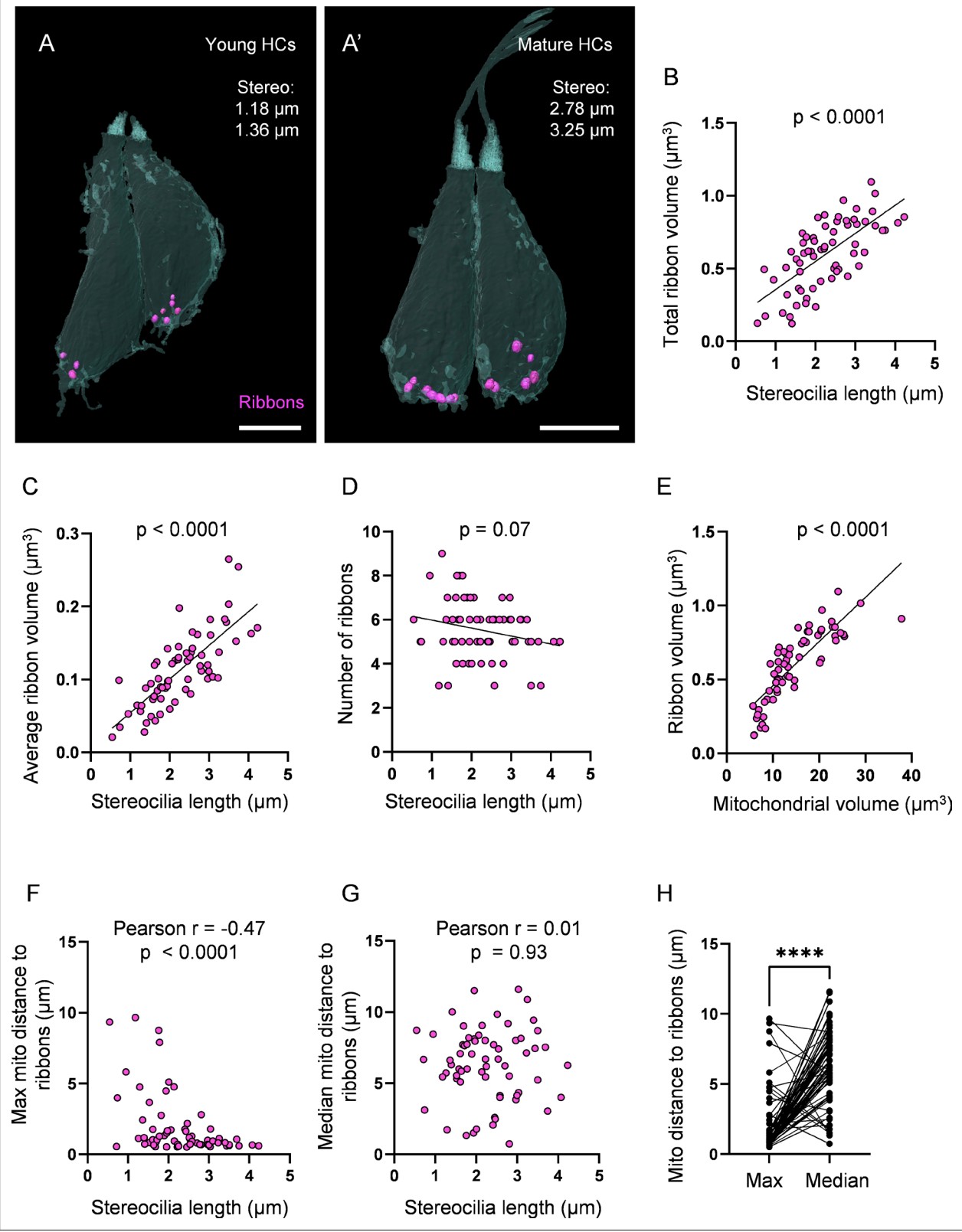

**Figure 3.** Ribbon growth parallels mitochondrial growth and localization. (**A**) Two representative young hair cells (HCs) from a 6 days post fertilization (dpf) neuromasts (NM) (NM4, *Figure 3—source data 2*) with synaptic ribbons shown in purple. Scale bar = 4.5 µm. (**A'**) Two representative mature HCs from a 6 dpf NM (NM4, *Figure 3—source data 2*). Scale bar = 4 µm. (**B**) Relationship between HC stereocilia length and the total ribbon volume. (**C**) Relationship between HC stereocilia length and the average ribbon volume. (**D**) Relationship between HC stereocilia length and the number of

*Figure 3 continued*

ribbons. (**E**) Relationship between HC total mitochondrial volume and HC ribbon volume. Black line = standard linear regression, with significance as indicated. (**F**) Relationship between HC stereocilia length and the average minimum distance between each ribbon and the max mito. (**G**) Relationship between HC stereocilia length and the average minimum distance between each ribbon and the median mito. (**H**) Average minimum distance between each ribbon and the HC max or median mito. (In μm) Max mito: 1.9 ± 0.3; median mito: 6.3 ± 0.3. Mann–Whitney test, ****$p<0.0001$. HC: n = 65, 5 NMs, 3 fish, 5–6 dpf.

The online version of this article includes the following source data and figure supplement(s) for figure 3:

**Source data 1.** Raw values used in *Figure 3*.

**Source data 2.** Datasets used in *Figure 3*.

**Figure supplement 1.** Measurement of hair cell (HC) ribbon number and volume.

**Figure supplement 1—source data 1.** Raw values used in *Figure 3—figure supplement 1*.

---

idea, stereocilia length of 3 dpf HCs were on average shorter than but fell within the range of those from 5 to 6 dpf HCs (*Figure 4—figure supplement 1*). Similar to 5–6 dpf HCs, 3 dpf HCs gained mitochondrial volume with increasing stereocilia length (*Figure 4A, A' and B*), with an average total mitochondrial volume of 14.4 ± 1.2 μm³ (*Figure 4B*). Cell volume and ratio of mitochondrial volume to cell volume were not different between 3 dpf HCs and 5–6 dpf HCs (*Figure 4C*). 3 dpf HCs had on average fewer mitochondria (24.9 ± 2 individual mitochondria) than 5–6 dpf HCs (*Figure 4D*). In tandem, the median mitochondrion volume in 3 dpf HCs (0.3 ± 0.02 μm³) remained larger than that of 5–6 dpf HCs, regardless of HC age (*Figure 4E*). Although the volume of the max mito on average was the same between 3 dpf and 5–6 dpf HCs, and the z-score was unaffected, it did not appear to significantly gain volume (*Figure 4F and G*). Additionally, the max mito did not localize to the basolateral pole, but remained randomly distributed throughout the HC (*Figure 4H–J*). 3 dpf HCs also had smaller ribbons (0.06 ± 0.007 μm³) and lower total ribbon volumes (0.35 ± 0.04 μm³) than 5–6 dpf HCs, though the number of ribbons (6 ± 0.4) was unaffected (*Figure 4—figure supplement 2A–C*). The localization of the max mito to ribbons at 3 dpf was less than at 5–6 dpf (*Figure 4—figure supplement 2E and F*), though the relationship between mitochondrial volume and ribbon volume remained unaltered (*Figure 4—figure supplement 2*). These data suggest that while the HC mitochondrial volume expands independent of HC biophysical properties, development of proper mitochondrial architecture follows maturation of the NM.

## Disrupting hair cell mitochondrial architecture impacts mitochondrial calcium buffering

To test the impact of mitochondrial architecture on mitochondrial function, we created a CRISPR mutant for the gene *opa1*, a conserved dynamin-like GTPase necessary for the fusion of the inner mitochondrial membrane for which loss of function results in mitochondrial fragmentation from yeast to humans (*Olichon et al., 2006*). The mutant was generated by introducing a 15 base pair insertion containing a premature stop codon 234 base pairs into the second exon, resulting in a truncated, nonfunctional protein ('Materials and methods'). As predicted, HCs of these mutants have highly fragmented mitochondria compared to wildtype (WT), a phenotype of complete penetrance readily observable by fluorescence microscopy (*Figure 5—figure supplement 1A'*). We confirmed differences in mitochondrial number by SBFSEM (five HCs, *Figure 5A*) and found a fourfold greater number of individual mitochondria (156 ± 23) compared to WT and no mitochondrial network equivalent to the max mito found in WT HCs (*Figure 5B*). The total mitochondrial volume, however, was unaffected (*Figure 5C*). We next measured mitochondrial calcium uptake during waterjet stimulation in *opa1* HCs. Fish were double transgenic for GCaMP3 targeted to the inner mitochondrial matrix, and RGECO expressed in the cytoplasm, both under an HC-specific promoter (*Tg(myo6:mitoGCaMP3;myo6:cytoRGECO)*, *Figure 5D and E*). A 20 s sinusoidal pressure wave of 10 Hz was applied to the HC bundle (see 'Materials and methods'; *Pickett et al., 2018*). Larvae were genotyped following calcium imaging. There was no difference in baseline HC mitochondrial calcium fluorescence between WT and *opa1* mutants (*Figure 5—figure supplement 1C*). There was also no difference in the cytoplasmic calcium response to waterjet between WT and *opa1* HCs (*Figure 5F and G*), in either peak amplitude (*Figure 5H*) or integrated area (*Figure 5I*), indicating that *opa1* HCs do not have severely impaired mechanotransduction. However, mitochondria from *opa1* HCs demonstrated a significantly

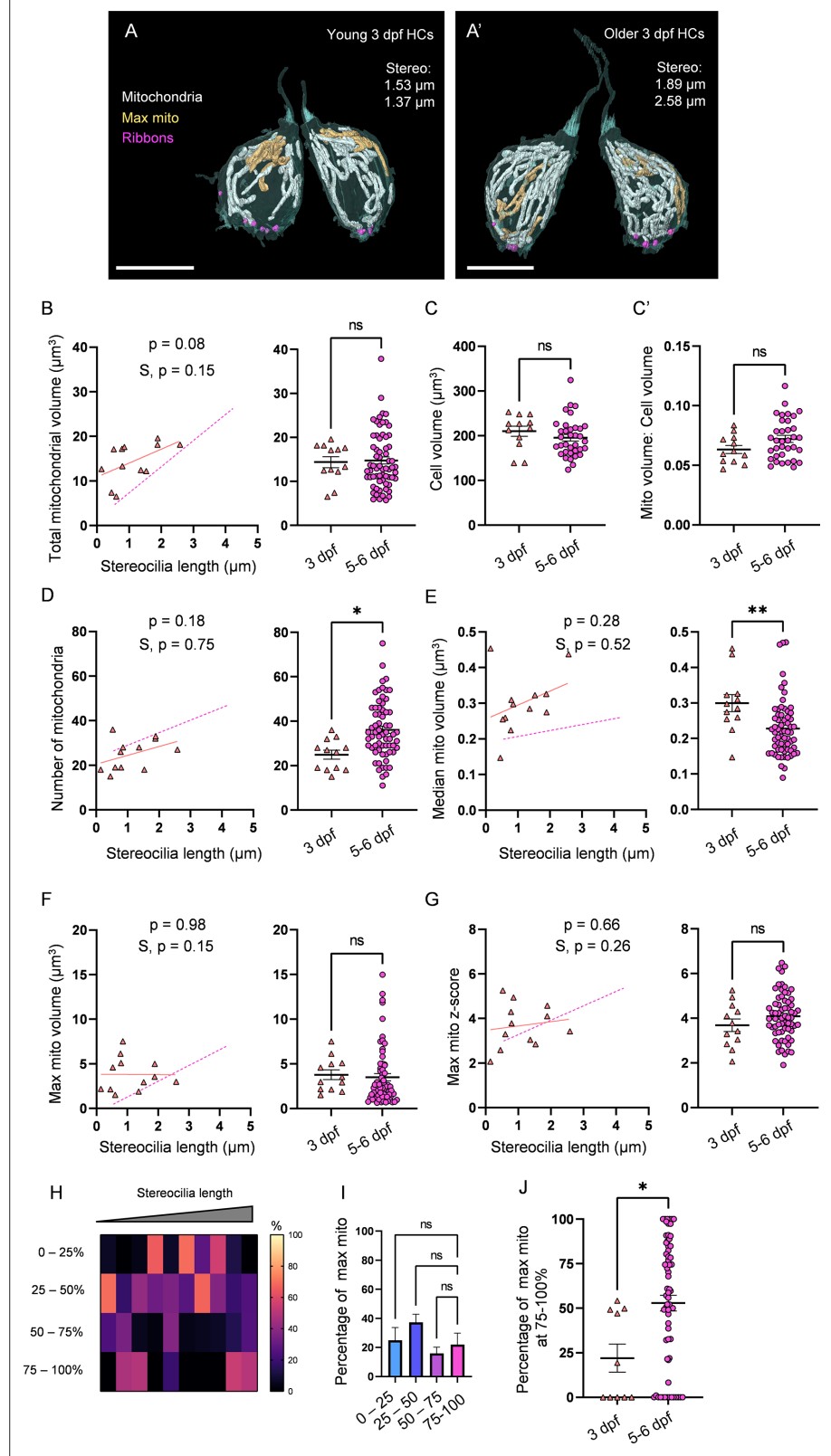

**Figure 4.** 3 days post fertilization (dpf) hair cells (HCs) demonstrate immature mitochondrial architecture. (**A**) Representative young HCs from a 3 dpf neuromasts (NM) (NM6, **Figure 4—source data 2**). Max mito shown in gold. Synaptic ribbons shown in purple. Scale bar = 8 μm. (**A'**) Representative older HCs from a 3 dpf NM (NM6, **Figure 4—source data 2**). Scale bar = 6 μm. (**B**) Comparison of total mitochondrial volume development (left)

*Figure 4 continued on next page*

*Figure 4 continued*

and average (right) between 3 dpf and 5–6 dpf HCs. On average (in µm³): 3 dpf: 14.4 ± 1.2; 5–6 dpf: 14.8 ± 0.8. Kolmogorov–Smirnov test, p=0.49. (**C**) Total HC volume for 3 dpf and 5–6 dpf HCs. (In µm³) 3 dpf: 210.1 ± 11.3; 5–6 dpf: 195.6 ± 7.2. Kolmogorov–Smirnov test, p=0.2. (**C′**) Ratio of total mitochondrial volume to HC volume. 3 dpf: 0.06 ± 0.003; 5–6 dpf: 0.07 ± 0.003. Kolmogorov–Smirnov test, p=0.41. (**D**) Comparison of the number of HC mitochondria over development (left) and on average (right) in 3 dpf and 5–6 dpf HCs. On average: 3 dpf: 25.0 ± 2; 5–6 dpf: 36.1 ± 1.6. Kolmogorov–Smirnov test, p=0.022. (**E**) Comparison of the median mitochondrial volume over development (left) and on average (right). On average (in µm³): 3 dpf: 0.3 ± 0.02; 5–6 dpf: 0.2 ± 0.01. Kolmogorov–Smirnov test, p=0.005. (**F**) Comparison of the max mito volume over development (left) and on average (right). On average (in µm³): 3 dpf: 3.8 ± 0.5; 5–6 dpf: 3.5 ± 0.4. Kolmogorov–Smirnov test, p=0.32. (**G**) Comparison of the max mito z-score in 3 dpf and 5–6 dpf HCs over development (left) and on average (right). On average: 3 dpf: 3.7 ± 0.3; 5–6 dpf: 4.1 ± 0.1. Standard unpaired *t*-test, p=0.21. (**B–G**) Solid line represents the standard linear regression for 3 dpf HCs. Dashed line represents standard regression for the 5–6 dpf HCs dataset as in *Figure 2*. Significance of the 3 dpf regression and difference with 5–6 dpf regression slope (S) are indicated. (**H**) The percentage of the max mito located within each quadrant of 3 dpf HCs represented as a heat map. Two HCs in the 3 dpf dataset lacked ribbons to provide a consistent HC lowest point and were not included in this analysis. (**I**) Summary of the heat map data shown in (**H**). Most apical quadrant (0–25%): 24.9 ± 8.9%; 25–50%: 37.2 ± 5.7%; 50–75%: 16.0 ± 4.2%; Most basal quadrant (75–100%): 22.0 ± 7.9%. Kruskal–Wallis test with Dunn's multiple comparisons, nonsignificant. (**J**) Percentage of max mito located within the most basal quadrant for individual HCs. 3 dpf: 22.0 ± 7.9%, 5–6 dpf: 52.8 ± 4.4%. Kolmogorov–Smirnov test, p=0.017. Same cells as in (**H, I**). (**B, D–G**) 3 dpf data: n = 12 HCs, 2 NMs, 2 fish. 5–6 dpf data: n = 65 HCs, 5 NMs, 3 fish. (**C, C′**) 3 dpf data: n = 12 HCs, 2 NMs, 2 fish. 5–6 dpf data: n = 35 HCs, 3 NMs, 3 fish. (**H–J**) 3 dpf data: n = 10 HCs, 2 NMs, 2 fish. 5–6 dpf data: n = 65 HCs, 5 NMs, 3 fish. Where applicable, data are presented as the mean ± SEM.

The online version of this article includes the following source data and figure supplement(s) for figure 4:

**Source data 1.** Raw values used in *Figure 4*.

**Source data 2.** Datasets used in *Figure 4*.

**Figure supplement 1.** 3 days post fertilization (dpf) hair cell (HC) stereocilia fall within the range of immature HCs.

**Figure supplement 1—source data 1.** Raw values used in *Figure 4*.

**Figure supplement 2.** 3 days post fertilization (dpf) hair cells (HCs) have smaller ribbons than 5–6 dpf HCs.

**Figure supplement 2—source data 1.** Raw values used in *Figure 4—figure supplement 2*.

reduced mitoGCaMP peak (*Figure 5J*) and integrated area (*Figure 5K*) during waterjet, indicating that they took up less calcium than WT siblings. Additionally, while WT HC mitochondria continue to sequester calcium beyond the termination of the waterjet stimulus (*Figure 5F*, also see *Pickett et al., 2018*), *opa1* mitochondria calcium levels returned to baseline in tandem with the stimulus cessation (*Figure 5G*). As fragmented mitochondria are associated with lower mitochondrial membrane potentials and decreased OXPHOS capacity, we used TMRE dye to measure mitochondrial membrane potential in *opa1* mutants (*Figure 5—figure supplement 1A and B*). HC mitochondria took up less of the TMRE dye in *opa1* mutants than WT, suggesting that opa1 HC mitochondria are depolarized. These results suggest that the development of networked mitochondria through fusion may preserve mitochondrial health, membrane potential, and capacity for calcium uptake during HC activity.

## Hair cell activity regulates the development of mitochondria architecture

We next asked how mechanotransduction shaped HC mitochondria. In many cell types, including skeletal muscle and cardiomyocytes, activity and intracellular calcium drive mitochondrial biogenesis to support an upregulated metabolic load (*Ojuka et al., 2002*; *Chin, 2004*). We hypothesized that development of mechanotransduction activity during HC maturation resulted in an increased energetic load that would similarly drive HC mitochondrial biogenesis and patterning. We reconstructed HCs and mitochondria from 5 dpf *cdh23* mutant zebrafish (*Figure 6A*, 2 fish, 4 NMs, 19 HCs total), which lack the tip-links necessary to open mechanotransduction channels (*Söllner et al., 2004*). Surprisingly, *cdh23* mutant HCs developed similar total mitochondrial volumes to WT (on average 14.6 ± 0.8 µm³; *Figure 6B*), and similarly mitochondria composed ~7% of the cell volume (*Figure 6C*). In contrast to WT, however, they had significantly fewer mitochondria (19.8 ± 1.4, *Figure 6D*). As a consequence, the volume of the median mitochondrion was larger in *cdh23* mutants than in WT (0.4

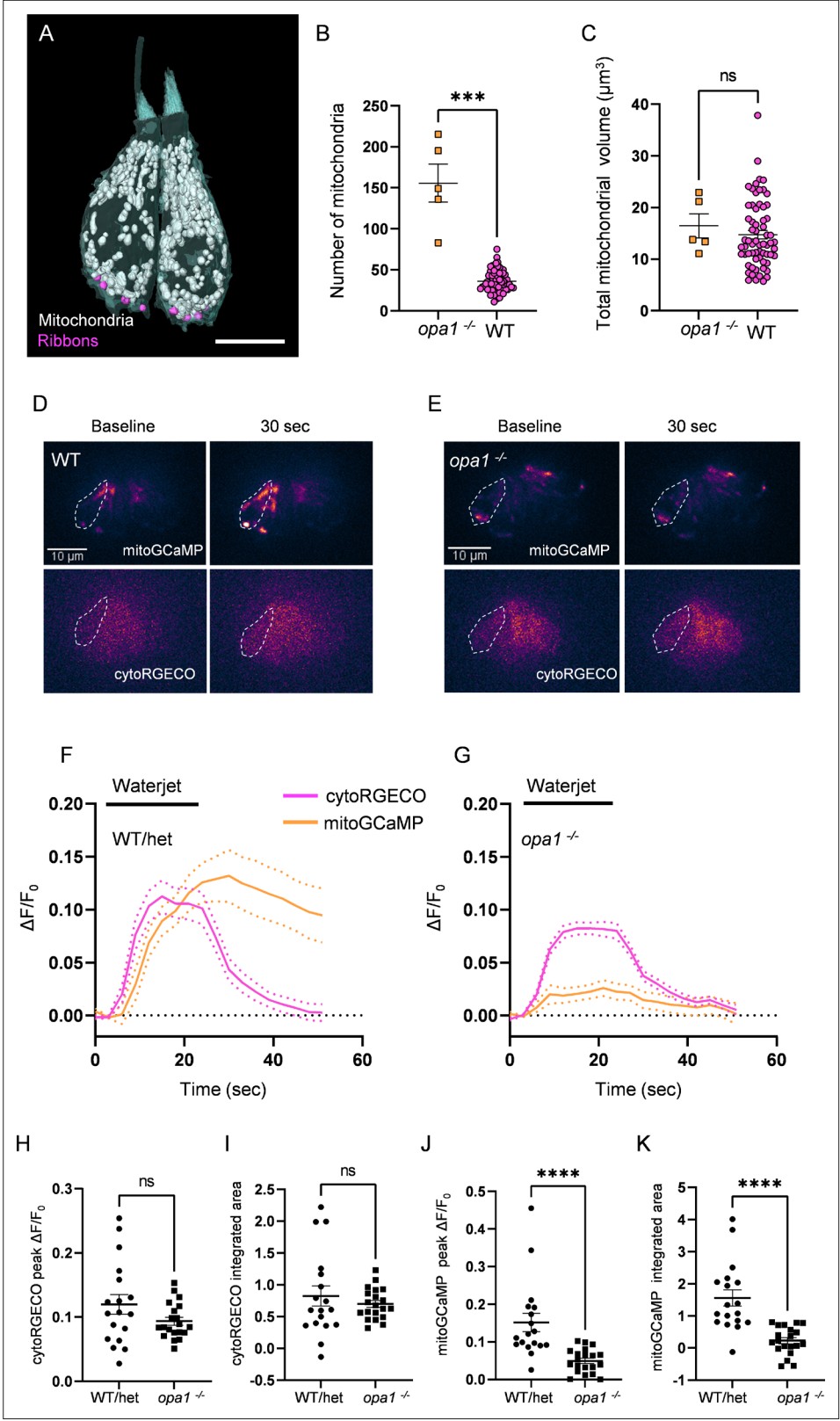

**Figure 5.** Mutations in *opa1* disrupt hair cell (HC) mitochondrial architecture and calcium buffering. (**A**) Two representative *opa1* HCs (*Figure 5—source data 2*). Scale bar = 5 µm. (**B**) Number of individual mitochondria in WT and *opa1* HCs. WT: 36.1 ± 1.6; *opa1*: 155.6 ± 23.22. Kolmogorov–Smirnov test, p=0.0002. (**C**) Total mitochondrial volume in WT and *opa1* HCs. In µm³, WT: 14.8 ± 0.8; *opa1*: 16.5 ± 2.3. Kolmogorov–Smirnov test,

*Figure 5 continued on next page*

*Figure 5 continued*

p=0.66. (**B, C**) *opa1* HCs; n = 5 HCs, 1 NM, 1 fish. (**D**) Representative WT images of *myo6:mitoGCaMP* (top) and *myo6:cytoRGECO* (bottom) at baseline and following a waterjet stimulus. (**E**) Same as in (**D**), but for *opa1* HCs. (**F, G**) Changes in *myo6:mitoGCaMP* and *myo6:cytoRGECO* $Ca^{2+}$ signal (expressed as $\Delta F/F_0$) following a 20 s, 10 Hz waterjet for both WT/het (**F**) and *opa1* HCs (**G**). (**H**) Peak *myo6:cytoRGECO* $\Delta F/F_0$ signal. WT/het: 0.1 ± 0.02; *opa1*: 0.1 ± 0.01, Mann–Whitney test, p=0.28. (**I**) Integrated *myo6:cytoRGECO* $\Delta F/F_0$ signal. WT/het: 0.8 ± 0.2; *opa1*: 0.7 ± 0.05, Mann–Whitney test, p=0.94. (**J**) Peak *myo6:mitoGCaMP* $\Delta F/F_0$ signal. WT/het: 0.15 ± 0.02; *opa1*: 0.05 ± 0.007, Mann–Whitney test, p<0.0001. (**K**) Integrated *myo6:mitoGCaMP* $\Delta F/F_0$ signal. WT/het: 1.6 ± 0.3; *opa1*: 0.2 ± 0.1, Mann–Whitney test, p<0.0001. (**D–K**) WT/het: n = 18 HCs, 7 fish. *opa1*: n = 20 HCs, 9 fish. Data are presented as the mean ± SEM.

The online version of this article includes the following source data and figure supplement(s) for figure 5:

**Source data 1.** Raw values used in *Figure 5*.

**Source data 2.** Datasets used in *Figure 5*.

**Figure supplement 1.** Mutations in *opa1* decrease TMRE uptake, but do not affect baseline mitochondrial calcium.

**Figure supplement 1—source data 1.** Raw values used in *Figure 5—figure supplement 1*.

---

± 0.04 $\mu m^3$, *Figure 6E*), regardless of HC age. While the max mito gained some volume over development (*Figure 6F*), the growth in raw volume was not significant, and it remained closer to the mean (average z-score, 3.0 ± 0.16, *Figure 6G*) than WT. Additionally, the max mito was randomly distributed throughout each HC without a strong preference for the basolateral quadrant (*Figure 6H–J*). Mutant HCs demonstrated a lower total ribbon volume (0.4 ± 0.04 $\mu m^3$) with a nonsignificant decrease in individual ribbon volume (0.08 ± 0.008 $\mu m^3$) compared with WT, but a significantly lower number of individual ribbons (4.6 ± 0.2, *Figure 6—figure supplement 1A–C*), although the correlation between mitochondrial volume and ribbon volume was preserved (*Figure 6—figure supplement 1D*). These results imply that while mechanotransduction is not necessary for the high HC mitochondrial volume, it is necessary for the nonuniform nature of the mature HC mitochondrial architecture, including the development of a max mito preferentially located in the base of the cell. In parallel, mechanotransduction activity is necessary for proper development of synaptic ribbons.

The consistent localization of the largest mitochondrion to the ribbons in WT HCs implies a role in synaptic transmission. We therefore next asked whether synaptic activity was necessary for max mito growth. We reconstructed HCs and their mitochondria from 5 dpf *cav1.3a* mutants (2 fish, 4 NMs, 48 HCs total, *Figure 7A*). Mutant HCs developed mitochondrial volume with age, indistinguishable from WT (*Figure 7B*). Like WT, *cav1.3a* mitochondria composed on average 7% of the total volume (*Figure 7C and C′*). Total mitochondrial number was also indistinguishable from WT, although they appeared to gain individual mitochondria faster than WT (*Figure 7D*). While the median mitochondrion in *cav1.3a* HCs was larger than WT across all ages (0.3 ± 0.02 $\mu m^3$, *Figure 7E*), it decreased in size over maturation (p=0.0009). In contrast to WT, the max mito in *cav1.3a* HCs did not change in volume during HC maturation (*Figure 7F*, p=0.35), and on average remained closer to the mean (*Figure 7G*, z-score, 3.4 ± 0.1) across all ages. The steady decrease in the size of the median mitochondrion resulted in a developmental increase in the z-score of the max mito, although it remained lower than WT (*Figure 7G*). Nevertheless, the max mito was preferentially found in the lowest quadrant of the HC (*Figure 7H and I*). Mutant HCs did not have any changes in total ribbon volume compared with WT (0.6 ± 0.03 $\mu m^3$; *Figure 7—figure supplement 1A*). However, there was a decrease in the average ribbon volume (0.08 ± 0.003 $\mu m^3$) and a subsequent increase in the number of individual ribbons (7.4 ± 0.3) compared with WT (*Figure 7—figure supplement 1B and C*). These ribbons were often found in tightly compact clusters (*Figure 7—figure supplement 1G*). Similar to WT, ribbon volume correlated positively with total mitochondrial volume (*Figure 7—figure supplement 1D*). Although closer than the median mito, the max mito did not become progressively localized to the ribbons (*Figure 7—figure supplement 1E and F*). Overall, these data imply that synaptic transmission is not required for the high mitochondrial volume or density. However, the absence of synaptic transmission might lead to changes in the mitochondrial architecture over maturation, with a steady decrease in the size of the median mitochondrion, no expansion of the max mito, nor progressive localization to the HC ribbons.

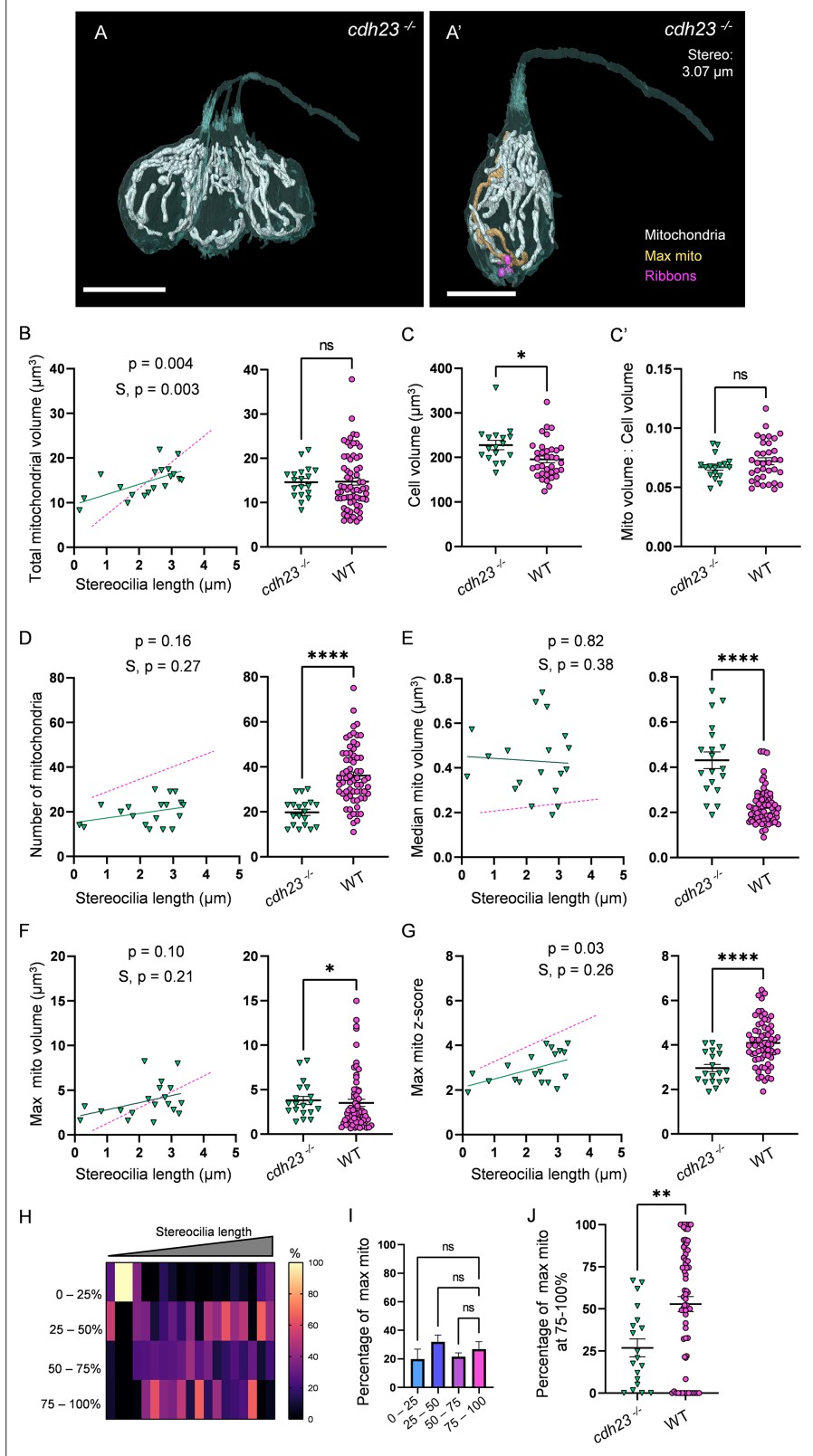

**Figure 6.** Mechanotransduction is not necessary for high mitochondrial volume, but is required for development of mitochondrial architecture. (**A**) Three representative *cdh23* hair cells (HCs) with mitochondria labeled in white (from NM11, *Figure 6—source data 2*). Scale bar = 9.5 µm. (**A'**) Representative mature *cdh23* HC (NM11), max mito shown in gold, synaptic ribbons shown in purple. Scale bar = 6 µm. (**B**) Comparison of total mitochondrial

*Figure 6 continued on next page*

*Figure 6 continued*

volume development (left) and average (right) between *cdh23* and WT HCs. On average (in μm³): *cdh23*:14.6 ± 0.8; WT: 14.8 ± 0.8. Kolmogorov–Smirnov test, p=0.35 (**C**) Total HC volume for *cdh23* and WT HCs. (In μm³) *cdh23*: 228.0 ± 10.3; WT: 195.6 ± 7.2. Kolmogorov–Smirnov test, p=0.03. (**C′**) Ratio of total mitochondrial volume to HC volume. *cdh23*: 0.07 ± 0.002; WT: 0.07 ± 0.003. Kolmogorov–Smirnov test, p=0.23. (**D**) Comparison of the number of HC mitochondria over development (left) and on average (right) in *cdh23* and WT HCs. On average: *cdh23*: 19.8 ± 1.4; WT: 36.1 ± 1.6. Kolmogorov–Smirnov test, p<0.0001. (**E**) Comparison of the median mitochondrial volume over development (left) and on average (right). On average (in μm³): *cdh23*: 0.4 ± 0.04; WT: 0.2 ± 0.01. Kolmogorov–Smirnov test, p<0.0001. (**F**) Comparison of the max mito volume over development (left) and on average (right). On average (in μm³): *cdh23*: 3.8 ± 0.5; WT: 3.5 ± 0.4. Kolmogorov–Smirnov test, p=0.04. (**G**) Comparison of the max mito z-score in *cdh23* and WT HCs over development (left) and on average (right). On average: *cdh23:* 3.0 ± 0.2; WT: 4.1 ± 0.1. Standard unpaired *t*-test, p<0.0001. (**B–G**) Solid line represents the standard linear regression for *cdh23* HCs. Dashed line represents standard regression for WT HCs as in *Figure 2*. Significance of the *cdh23* regression and differences in the slope from the WT regression (S) are indicated. (**H**) The percentage of the max mito located within each quadrant of *cdh23* HCs represented as a heat map. (**I**) Summary of the heat map data shown in (**H**). Most apical quadrant (0–25%): 19.8 ± 7.0%; 25–50%: 31.9 ± 4.8%; 50–75%: 21.6 ± 2.6%; Most basal quadrant (75–100%): 26.8 ± 5.3%. Kruskal–Wallis test with Dunn's multiple comparisons, nonsignificant. (**J**) Percentage of max mito located within the most basal quadrant for individual HCs. *cdh23*: 26.8 ± 5.3%, WT: 52.8 ± 4.4%. Kolmogorov–Smirnov test, p=0.008. (**B, D–J**) *cdh23* data: n = 19 HCs, 4 NMs, 2 fish. WT data: n = 65 HCs, 5 NMs, 3 fish. (**C, C′**) *cdh23* data: n = 17 HCs, 4 NMs, 2 fish. WT data: n = 35 HCs, 3 NMs, 3 fish. Where applicable, data are presented as the mean ± SEM.

The online version of this article includes the following source data and figure supplement(s) for figure 6:

**Source data 1.** Raw values used in *Figure 6*.

**Source data 2.** Datasets used in *Figure 6*.

**Figure supplement 1.** Mechanotransduction-deficient hair cells (HCs) have fewer ribbons than WT.

**Figure supplement 1—source data 1.** Raw values used in *Figure 6—figure supplement 1*.

## Multidimensional analysis of HC mitochondrial properties confirms differences across genotypes

For a more comprehensive comparison, we wanted to determine whether there was still a significant difference between WT and mutant HCs when comparing all of the measured mitochondrial features. However, these measurements likely have co-variance, complicating the analysis. Principal component analysis (PCA) is a useful approach for this type of comparison where multiple variables may be correlated as the principal components are themselves by nature uncorrelated. We then used uniform manifold approximation and projection (UMAP) to represent the multidimensional PCA space as a projection in two dimensions. We performed PCA with eight aspects of HC mitochondria: (1) number of mitochondria, (2) the total mitochondrial volume, (3) volume of the max mito, (4) volume of the median mitochondrion, (5) z-score of max mito, (6) max mito cable length, (7) average minimum distance of max mito to the ribbons, and (8) average minimum distance of the median mitochondrion to the ribbons. We used the first six principal components, which described 97.8% of the variability in the dataset, for UMAP representation (*Figure 8*).

We first analyzed wildtype HCs, comparing their spatial distribution with cell maturity as measured by stereocilia length. We color-coded HCs according to the length of their stereocilium, finding a gradual transition between young and mature HC mitochondrial properties (*Figure 8A*). We performed spatial autocorrelation with stereocilium length and found a significant correlation across the manifold (Moran's I = 0.3, p=0.001). We also observed a separation of mitochondrial properties across all three animal ages (3–6 dpf, *Figure 8B*). As predicted by our prior analysis (*Figure 4*), we see a mixture of HCs from different age animals since at each age there is a range of HC maturity. These results confirm a trajectory of HC mitochondrial maturation across the UMAP projection, consistent with a gradual development of this phenotype, and show that when multiple aspects of HC mitochondria are taken together, older HCs have a significantly different phenotype than younger HCs.

To evaluate how both mechanotransduction and synaptic activity regulate the development of the HC mitochondrial phenotype, we included both mutants in the PCA (*Figure 8C and D*). We first confirmed that mutants showed a relationship between stereocilia length and kinocilium length similar to WT (*Figure 8—figure supplement 1*), suggesting these metrics are independent of changes in

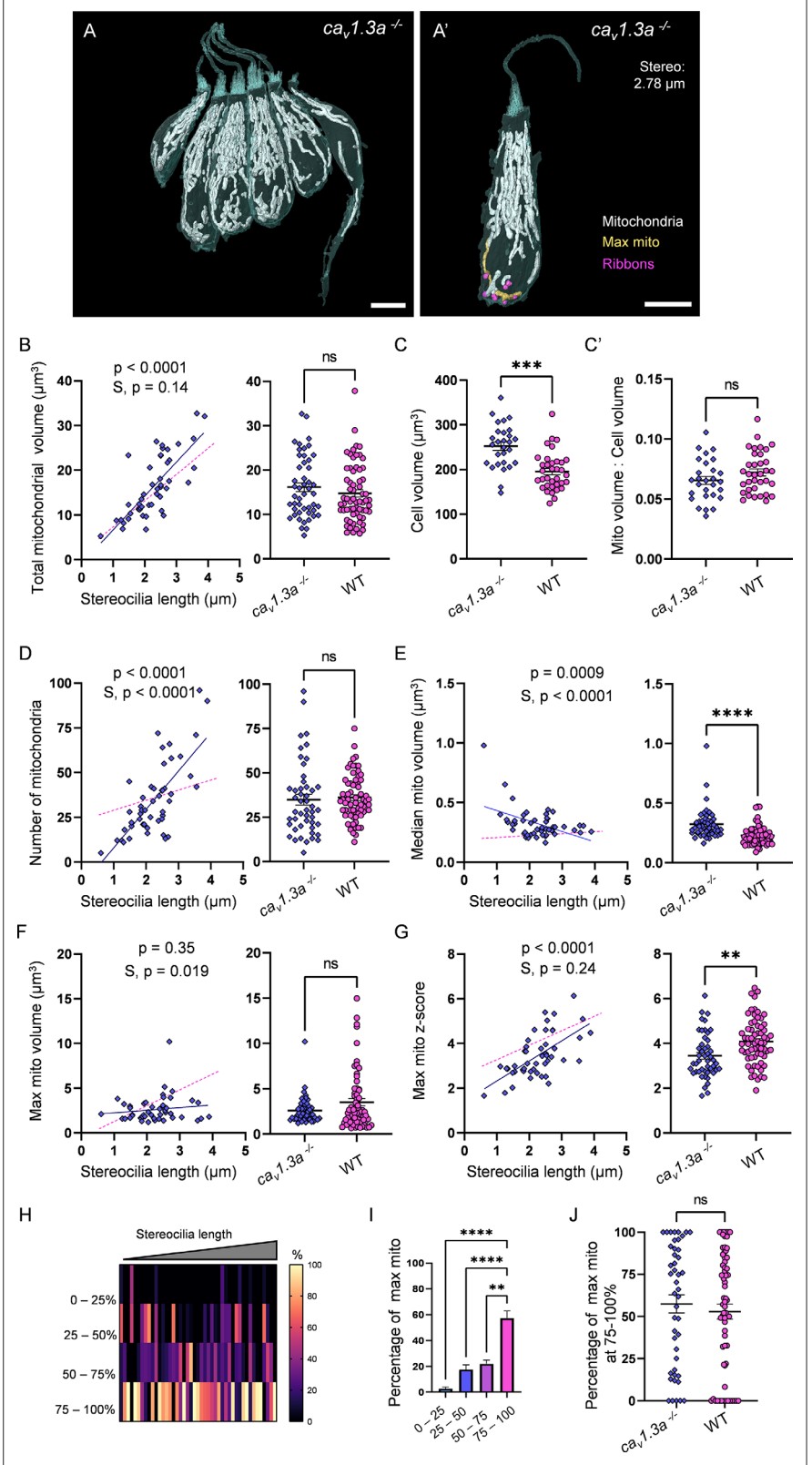

**Figure 7.** Synaptic transmission is necessary for gradual max mito growth. (**A**) Six representative *cav1.3a* hair cells (HCs) with mitochondria labeled in white (from NM13, *Figure 7—source data 2*). Scale bar = 5 μm. (**A'**) Representative mature *cav1.3a* HC (NM14, *Figure 7—source data 2*), max mito shown in gold, synaptic ribbons shown in purple. Scale bar = 5 μm. (**B**) Comparison of total mitochondrial volume development (left)

*Figure 7 continued on next page*

*Figure 7 continued*

and average (right) between *cav1.3a* and WT HCs. On average (in µm³): *cav1.3a*: 16.2 ± 1.0; WT: 14.8 ± 0.8. Kolmogorov–Smirnov test, p=0.76. (**C**) Total HC volume for *cav1.3a* and WT HCs. (In µm³) *cav1.3a*: 252.5 ± 9.3; WT: 195.6 ± 7.2. Kolmogorov–Smirnov test, p=0.0003. (**C′**) Ratio of total mitochondrial volume to HC volume. *cav1.3a*: 0.07 ± 0.003; WT: 0.07 ± 0.003. Kolmogorov–Smirnov test, p=0.66. (**D**) Comparison of the number of HC mitochondria over development (left) and on average (right) in *cav1.3a* and WT HCs. On average: *cav1.3a*: 34.9 ± 3.0; WT: 36.1 ± 1.6. Kolmogorov–Smirnov test, p=0.13. (**E**) Comparison of the median mitochondrial volume over development (left) and on average (right). On average (in µm³): *cav1.3a*: 0.3 ± 0.02; WT: 0.2 ± 0.01. Kolmogorov–Smirnov test, p<0.0001. (**F**) Comparison of the max mito volume over development (left) and on average (right). On average (in µm³): *cav1.3a*: 2.6 ± 0.2; WT: 3.5 ± 0.4. Kolmogorov–Smirnov test, p=0.14. (**G**) Comparison of the max mito z-score in *cav1.3a* and WT HCs over development (left) and on average (right). On average: *cav1.3a*: 3.4 ± 0.1; WT: 4.1 ± 0.1. Standard unpaired *t*-test, p=0.001. (**B–G**) Solid line represents the standard linear regression for *cav1.3a* HCs. Dashed line represents standard regression for WT HCs dataset as in *Figure 2*. Significance of the *cav1.3a* regression and differences in the slope from the WT regression (**S**) are indicated. (**H**) The percentage of the max mito located within each quadrant of *cav1.3a* HCs represented as a heat map. Three HCs in the *cav1.3a* dataset lacked ribbons to provide a consistent HC lowest point and were not included in this analysis. (**I**) Summary of the heat map data shown in (**H**). Most apical quadrant (0–25%): 2.8 ± 1.2%; 25–50%: 17.6 ± 3.6%; 50–75%: 22.1 ± 2.8%; Most basal quadrant (75–100%): 57.5 ± 5.4%. Kruskal–Wallis test with Dunn's multiple comparisons, **p<0.01, ****p<0.0001. (**J**) Percentage of max mito located within the most basal quadrant for individual HCs. *cav1.3a*: 57.4 ± 5.4%, WT: 52.8 ± 4.4%. Kolmogorov–Smirnov test, p=0.77. (**B, D–G**) *cav1.3a* data: n = 48 HCs, 4 NMs, 2 fish. WT dpf data: n = 65 HCs, 5 NMs, 3 fish. (**C, C′**) *cav1.3a* data n = 28 HCs, 3 NMs, 2 fish. WT data: n = 35 HCs, 3 NMs, 3 fish. (H - J) *cav1.3a* data: n = 45 HCs, 4 NMs, 2 fish. WT data: n = 65 HCs, 5 NMs, 3 fish. Where applicable, data are presented as the mean ± SEM.

The online version of this article includes the following source data and figure supplement(s) for figure 7:

**Source data 1.** Raw values used in *Figure 7*.

**Source data 2.** Datasets used in *Figure 7*.

**Figure supplement 1.** Synaptic transmission-deficient hair cells (HCs) have a larger number of smaller ribbons than WT.

**Figure supplement 1—source data 1.** Raw values used in *Figure 7—figure supplement 1*.

activity. In *Figure 8C and D*, WT data were compared to either *cdh23* or *cav1.3a* data, respectively. In each distribution, stereocilia length showed positive spatial autocorrelation (C, Moran's I = 0.3, p=0.02; D, Moran's I = 0.4, p=0.0001). Nearest-neighbor analysis demonstrated that both *cdh23* and *cav1.3a* mutant HCs were more likely to cluster with themselves (p=0.001, both *cdh23* and *cav1.3a*) than with WT (p=1.0 both *cdh23* and *cav1.3a*). Conclusions were unchanged when omitting WT data from 3 dpf NMs (*Figure 8—figure supplement 2*). These analyses support our findings that mutations in either mechanotransduction or synaptic transmission disrupt proper development of the hair mitochondrial phenotype.

## Discussion

Although mitochondria have long been known to be essential for HC function, a three-dimensional ultrastructural understanding of HC mitochondrial architecture, and its relationship to cell function, remained understudied. We used SBFSEM to describe an HC-specific mitochondrial phenotype characterized by (1) high mitochondrial volume and (2) a particular mitochondrial architecture, consisting of small mitochondria apically and large, networked mitochondria (max mito) near the synaptic ribbons. There are several caveats to using SBFSEM to study mitochondrial architecture. Although it is essential for resolving individual mitochondria, SBF requires fixed tissue, providing only a snapshot of the live mitochondrial dynamics that underlie the observed architecture. In addition, the laborious nature of the reconstructions precludes multiple replicates. However, we found no significant differences between HCs from different NMs and from different 5–6 dpf WT fish (65 HCs, 5 NMs from 3 fish), giving us confidence that our data reflect the greater population.

We found that lateral line HCs contain about 40 mitochondria in total. While this is over twice the number found in the surrounding SCs, it is notably on the smaller scale compared with numbers reported for other cell types, where estimates range from hundreds to thousands (*Robin and Wong, 1988*; *Smith and Ord, 1983*). We note these estimates were made from micrographs or measurements

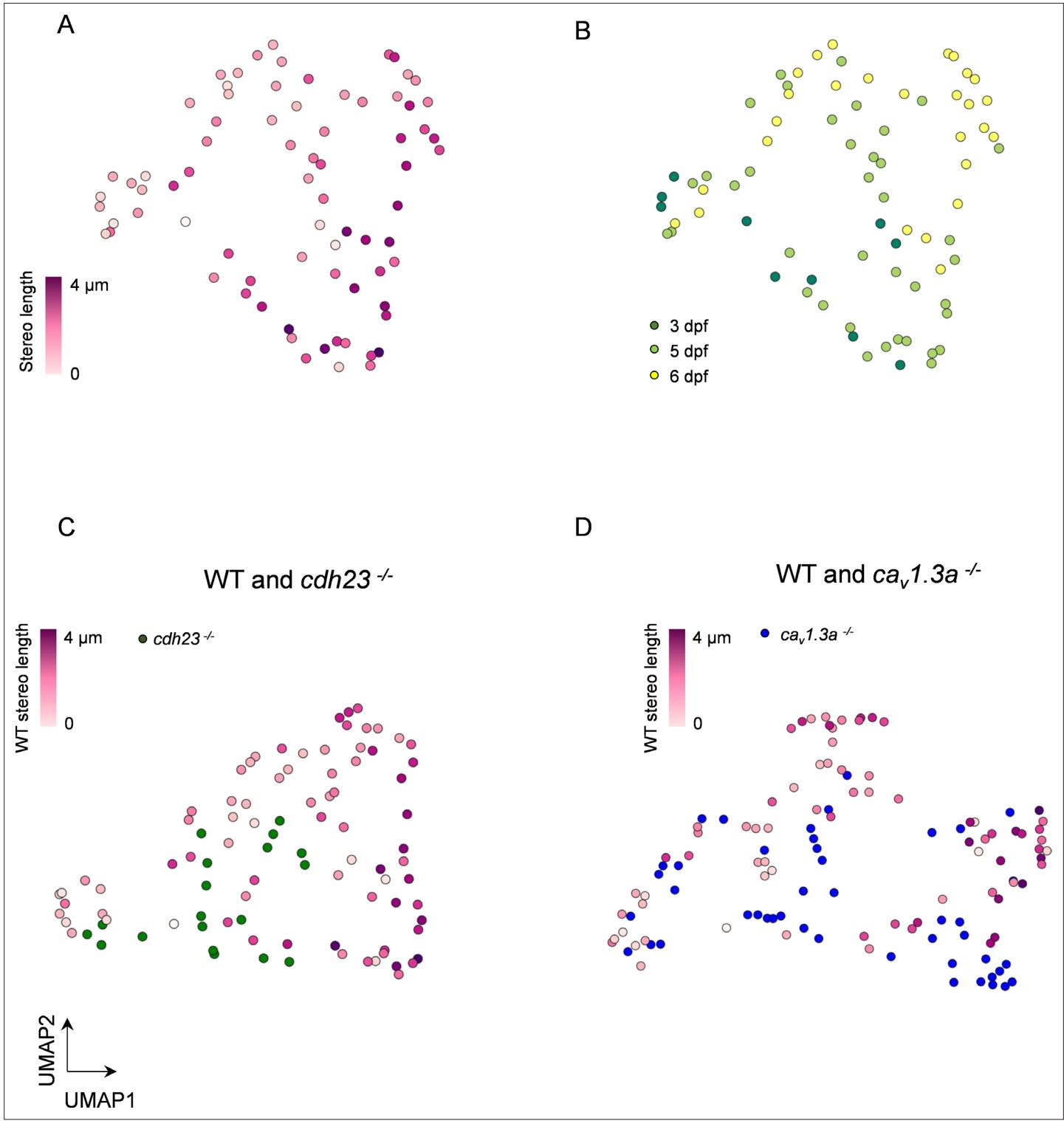

**Figure 8.** Multidimensional analysis of hair cell (HC) mitochondrial properties confirms differences across genotypes. (**A**) UMAP plot of HCs based on principal component analysis of mitochondrial properties. Variables included in this analysis: (1) number of mitochondria, (2) total mitochondrial volume, (3) max mito volume, (4) max mito cable length, (5) median mito volume, (6) average minimum distance between max mito and ribbons, and (7) average minimum distance between median mito and ribbons. HCs are color-coded by stereocilia length. (**B**) Same UMAP analysis as in (**A**), color-coded according to age of the animal. Total HCs, n = 75. 3 days post fertilization (dpf) HC: n = 10, 2 NM, 2 fish. 5 dpf HCs: 38 HC, 3 NM, 2 fish. 6 dpf HC: n = 27 HC, 2 NM, 1 fish. (**C**) UMAP analysis comparing WT to *cdh23* HCs. (**D**) UMAP plot comparing WT to *cav1.3a* HC. WT HC data, n = 75 HCs, 7 NM, 5 fish (3–6 dpf, same as **A** and **B**). *cdh23* data, n = 19, 4 NM, 2 fish, 5 dpf. *cav1.3a* data, n = 45, 4 NM, 2 fish, 5 dpf.

*Figure 8 continued on next page*

*Figure 8 continued*

The online version of this article includes the following source data and figure supplement(s) for figure 8:

**Source data 1.** Raw values used in *Figure 8*.

**Figure supplement 1.** *cav1.3a* and *cdh23* have similar relationships between stereocilia and kinocilia lengths as WT.

**Figure supplement 1—source data 1.** Raw values used in *Figure 8—figure supplement 1*.

**Figure supplement 2.** 3 days post fertilization (dpf) hair cell (HC) data does not affect UMAP separation.

**Figure supplement 2—source data 1.** Raw data used in *Figure 8—figure supplement 2*.

of mtDNA, which can vary widely. There are few studies that have similar complete reconstructions of total mitochondria in cells using high-resolution methodologies. A recent study indicates that there are nearly 500 mitochondria found in a primate cone photoreceptor after SBFSEM reconstruction (*Hayes et al., 2021*). Another recent study *Liu et al., 2022* used SBFSEM to reconstruct mitochondria from mammalian cochlear HCs and estimated mitochondria numbered into the thousands. A better established measurement for comparison is mitochondrial volume as a fraction of the total cell volume (*Posakony et al., 1977*). Our measured ratio of mitochondrial volume to total cell volume in HCs (~7%, *Figure 1H*) is consistent with another recent study of mouse outer HCs that used EM tomography and estimated mitochondrial volume about 10% of cytoplasmic volume (*Perkins et al., 2020*). These numbers are also in alignment with SBFSEM reconstruction of neurons and glia in rat cortex, with mitochondrial volumes to cell volumes of 6–10% (*Calì et al., 2019*), though unfortunately mitochondrial number was not reported in this study.

We find distinct mitochondrial architecture along the HC apicobasal axis. At the HC apical pole, we find smaller mitochondria. In rat cochlear HCs, apical mitochondria take up calcium during mechanotransduction (*Beurg et al., 2010*), contributing to robust calcium buffering mechanics to maintain cytoplasmic calcium concentrations that could otherwise affect mechanotransduction and adaptation (*Ricci et al., 1998*; *Eatock et al., 1987*). Calcium influx itself could also lead to the smaller size of apical mitochondria. In neurons, calcium has been reported to induce mitochondrial fission (*Rintoul et al., 2003*), and the fission regulator Drp1 is activated by calcium (*Cribbs and Strack, 2007*; *Han et al., 2008*; *Cereghetti et al., 2008*). As excess calcium uptake can lead to mitochondrial damage (*Starkov et al., 2004*), mitochondrial fission in this region might facilitate mitophagy and quality control (*Pfluger et al., 2015*; *Twig et al., 2008*). The small size may also provide increased efficiency in packing, helping to maintain distinct calcium pools in apical and basal compartments, as has been suggested for photoreceptor mitochondria (*Giarmarco et al., 2017*). Mitochondria in *opa1* mutants, which are uniformly small, demonstrated reduced calcium buffering. This could be a result of their small size or diminished mitochondrial health from lack of fusion, preventing resource sharing and hence leading to depolarized potentials. However, we note that mutations in *opa1* will have multiple effects on mitochondrial function in addition to changes in size that may also influence their ability to take up calcium.

At the basolateral pole, HCs contained large mitochondrial networks (max mito) associated with ribbon synapses. Synaptic transmission requires large energetic expenditures (*Li and Sheng, 2022*). The proximity of the networked mitochondria to the synaptic ribbons suggests these large mitochondria might serve as active metabolic support. Mitochondrial fusion boosts oxidative phosphorylation and sharing of mitochondrial resources (*Picard et al., 2013*), indicating that the mitochondrial networks we observe would be particularly well-suited for this purpose. Synaptic mitochondria have been shown to take up calcium that enters through L-type CaV1.3 calcium channels, and disrupting this mitochondrial calcium uptake deregulated synaptic transmission (*Wong et al., 2019*). Larger mitochondria also have greater capacity to take up calcium (*Kowaltowski et al., 2019*; *Szabadkai et al., 2006*). As mitochondrial calcium uptake stimulates ATP production, interactions between mitochondrial fusion and synapse activity would be well-tuned to the cell's energetic demands.

Our findings show that neither proper mechanotransduction nor synaptic activity is necessary for the growth in mitochondrial volume during HC maturation. This is surprising, given that HCs rely on oxidative phosphorylation for 75% of their metabolic needs (*Puschner and Schacht, 1997*), and changes in metabolic activity is a primary driver for mitochondrial biogenesis in many tissues (*Jornayvaz and Shulman, 2010*). Moreover, the role of calcium influx stimulating mitochondrial biogenesis in many other electrically active cell types, such as skeletal muscle and cardiomyocytes,

is well established (*Ojuka et al., 2002*; *Chin, 2004*). We note that *cdh23* mutations used to disrupt mechanotransduction still exhibit low levels of spontaneous synaptic release (*Trapani and Nicolson, 2011*) and cannot rule out the possibility that this residual activity might be sufficient to promote biogenesis. However, we can conclude that the growth of the mitochondrial volume is not regulated by overall activity levels. Additionally, we found that disrupting the mitochondrial architecture with a mutation in *opa1* has no effect on the total mitochondrial volume. These fragmented mitochondria demonstrated depolarized potentials and decreased calcium uptake, likely associated with reduced ATP production. Together, these results suggest that the developmental increase in mitochondrial biogenesis is robust to changes in metabolic demands.

As the youngest HCs have a total mitochondrial volume similar to that of peripheral SCs, which serve as HC progenitors (*Thomas and Raible, 2019*), we suggest that the increase in mitochondrial volume is not linked to initial cell fate specification. Supporting this, the total mitochondrial volume increases gradually as HCs mature. This developmental mitochondrial growth is not driven by changes in cell volume, as described in other cell types (*Rafelski et al., 2012*; *Miettinen and Björklund, 2017*). Instead, it must be a product of different, ongoing pathways. One possibility may include the sirtuin deacetylases (SIRTs), key sensors of metabolism (*Nogueiras et al., 2012*) that can upregulate mitochondrial biogenesis through a series of parallel pathways (*Yuan et al., 2016*). In support of this, SIRT1 has been shown to be highly expressed in cochlear inner ear HCs (*Xiong et al., 2014*), and upregulation of SIRT1 pathways is protective against various modes of HC death (*Zhan et al., 2021*; *Liang et al., 2021*). The steady expansion of mitochondrial volume suggests these biogenesis-promoting pathways outweigh mitophagy and quality control. Such high mitochondrial volumes may produce dangerous reactive oxygen species (ROS) levels as a by-product of oxidative phosphorylation (*Zhu et al., 2013*). Therefore, the high mitochondrial volume might render HCs vulnerable to outside stresses that would additionally increase intracellular ROS. The observed mitochondrial networks might also in part counteract this vulnerability. Promoting mitochondrial fusion is protective against starvation-mediated apoptosis (*Gomes et al., 2011*) and SIRTs also regulate mitochondrial fusion (*Uddin et al., 2021*). SIRT3, located to mitochondria, has been implicated in counteracting age-related hearing loss and noise-induced damage (*Someya and Prolla, 2010*; *Brown et al., 2014*; *Patel et al., 2020*).

We found that mechanotransduction, while having little effect on mitochondrial biogenesis, was necessary for mitochondrial architecture. The clustered position of mechanotransduction mutants about midway within the mitochondrial UMAP trajectory (*Figure 8*) suggests a state of incomplete development where cells are unable to progress further. Lack of mechanotransduction resulted in fewer, larger, uniformly sized mitochondria. This is consistent with a model where apical calcium through mechanotransduction channels promotes mitochondrial fission. Mitochondria in mechanotransduction mutants also did not form basolateral networks localizing to synaptic ribbons. The fact that interference with mechanotransduction has effects on both formation of small mitochondria and localization of large, single networks suggests complex regulation of mitochondrial morphology.

By contrast, basal calcium entry associated with synaptic transmission is necessary for progressive growth of basal mitochondrial networks, but not formation of smaller apical mitochondria, which rapidly grew in number in *cav1.3a* mutants. In UMAP space, *cav1.3a* mutants intersperse with younger WT HCs, but segregate from mature HCs (*Figure 8*). Previous work (*Trapani and Nicolson, 2011*) has shown that *cav1.3a* mutants demonstrate reduced microphonic potentials, reflecting decreased mechanotransduction. If calcium influx drives apical mitochondrial fission, it is present at a level enough to do so in these mutants. Basal mitochondria take up calcium during synaptic transmission (*Wong et al., 2019*). This suggests a paradox where calcium might promote both mitochondrial fission apically and fusion basally. However, synaptic transmission requires large energy expenditures in addition to calcium regulation, and these demands may instead drive mitochondrial fusion into basal networks. Understanding the paradoxical effects of HC activity influencing both the formation of smaller apical mitochondria and larger basal networks will require additional study.

We find that altering calcium entry and synaptic transmission also altered ribbon size and morphology. These results compare with previous work showing that *cav1.3a* mutant HCs have enlarged ribbons (*Sheets et al., 2012*). *Wong et al., 2019* demonstrated that blocking mitochondrial calcium entry also resulted in larger ribbons, a phenotype mimicked by altering NAD⁺/NADH ratios. While both of these studies showed larger ribbons after manipulating calcium and mitochondrial

function, we observed *cav1.3a* mutant ribbons to be on average smaller than WT, but frequently found in clusters. We believe this discrepancy might be accounted for by the fact that the smaller ribbons we found in clusters at the EM level would appear to be larger single ribbons using fluorescence light microscopy methods employed in these previous studies. We also used strict structural parameters to define ribbons, which might account for differences in observed ribbon numbers and their locations as seen in fluorescence (*Wong et al., 2019*).

Not all HCs within zebrafish NMs are synaptically active, with active HCs tending to be younger cells on the NM periphery (*Zhang et al., 2018*; *Wong et al., 2019*; *Lukasz et al., 2022*). This is interesting, considering that we find the largest mitochondrial networks in more mature HCs. This discrepancy might be explained by considering that the initial onset of calcium entry associated with synaptic transmission might be sufficient to drive mitochondrial fusion, and that basolateral mitochondria remain networked regardless of whether the HC returns to a synaptically silent state during maturation.

Overall, our study provides a high-resolution, three-dimensional picture of zebrafish HC mitochondria, and demonstrates that through mechanotransduction and synaptic activity, these cells develop a finely tuned mitochondrial phenotype reflective of their function. This HC phenotype, through its high mitochondrial volume and large, appropriately positioned networked mitochondria, might be necessary to support high metabolic demands, but could also increase vulnerability to small fluctuations in intracellular ROS. Disruption of this phenotype could lead to improper HC physiology. Thus, it is critical that future studies take into account the high specificity with which HCs regulate their mitochondria.

# Materials and methods

## Key resources table

| Reagent type (species) or resource | Designation | Source or reference | Identifiers | Additional information |
|---|---|---|---|---|
| Strain, strain background (*Danio rerio*) | Opa1 mutant | This paper | ZFIN: ZDB-ALT-240524-3 | *opa1*^w264 mutant |
| Sequence-based reagent | Opa1 guide | This paper | http://crisperscan.org | 5′ GGCGAGACGGGCCACCCAGA 3′ (IDT) |
| Sequence-based reagent | Opa1 guide | This paper | http://crisperscan.org | 5′ GGCAGTGAGGTGGTCTCTGT 3′ (IDT) |
| Sequence-based reagent | Opa1 fwd primer | This paper | PCR primers | 5′ GCTGCCCGGCATTACACATCTC 3′ (IDT) |
| Sequence-based reagent | Opa1 rev primer | This paper | PCR primers | 5′ GCTCAGCGGTTGGAGGTGGATA 3′ (IDT) |
| Strain, strain background (*D. rerio*) | Cav1.3a mutant | PMID: 15115817 | ZDB-GENE-030616-135 | *cav1.3a*^tc323d mutant |
| Strain, strain background (*D. rerio*) | Cdh23 mutant | PMID: 9491988 | ZDB-GENE-040413-7 | *cdh23*^tj264 mutant |
| Strain, strain background (*D. rerio*) | mitoGCaMP3 | PMID: 25031409 | ZDB-TGCONSTRCT-141008-1 | Tg(myosin6b:mitoGCaMP3)^w119 |
| Strain, strain background (*D. rerio*) | cytoRGECO | PMID: 25114259 | ZDB-TGCONSTRCT-150114-2 | Tg(myosin6b:R-GECO1)^vo10Tg |
| Strain, strain background (*D. rerio*) | mitoGFP | This paper | ZFIN: ZDB-ALT-240529-8 | Tg(myosin6b:mitoGFP)^w213aTg; Gateway cloning and Tol2-mediated transgenesis |

*Continued on next page*

*Continued*

| Reagent type (species) or resource | Designation | Source or reference | Identifiers | Additional information |
|---|---|---|---|---|
| Chemical compound, drug | TMRE | Invitrogen | Invitrogen: T669 | |
| Software, algorithm | GraphPad Prism | GraphPad Software | https://www.graphpad.com | |
| Software, algorithm | Slidebook | Intelligent Imaging Innovations (3i) | https://www.intelligent-imaging.com | |
| Software, algorithm | Fiji | PMID: 22743772 | https://fiji.sc | |
| Software, algorithm | TrakEM2 | PMID: 22723842 | https://imagej.net/plugins/trakem2/ | |
| Software, algorithm | AMIRA 6.5 for EM systems | Thermo Fisher Scientific | http://Thermofisher.com | |

## Zebrafish

All experiments were done in compliance with the University of Washington Institutional Animal Use and Care Committee (IACUC protocol number 2997-01). Experiments were conducted when fish were 5 dpf unless otherwise noted. Sex is not determined at this age. Larvae were raised in embryo medium (EM, 14.97 mM NaCl, 500 µM KCl, 42 µM Na$_2$HPO4, 150 µM KH$_2$PO$_4$, 1 mM CaCl$_2$ dehydrate, 1 mM MgSO$_4$, 0.714 mM NaHCO$_3$, pH 7.2) at 28.0°C.

## Genetics lines

WT animals were of the AB strain. Both *cdh23*[tj264] (ZDB-GENE-040413–7) and *cav1.3a*[tc323d] (also known as *cacna1da*, ZDB-GENE-030616-135) mutant lines were generously received from the Nicolson lab (Stanford, Palo Alto, CA). Tg[*myo6b:mitoGcaMP*][w119] fish express GcaMP3 targeted to the mitochondrial matrix (*Esterberg et al., 2014*; *Pickett et al., 2018*). Both *cdh23*[tj264] and *cav1.3a*[tc323d] mutants could be identified by phenotype as these fish lie on their sides and do not demonstrate proper righting behavior.

The *opa1* mutant (*opa1*[w264]) was generated through CRISPR Cas9 technology. CRISPR guides were designed via http://crisperscan.org targeting *opa1* (mitochondrial dynamin like GTPase, ZDB-GENE-041114-7). Two gRNAs were prepared, with sequences GGCGAGACGGGCCACCCAGA and GGCA GTGAGGTGGTCTCTGT. Both guides targeted the second exon. The gRNAs were prepared according to the protocol outlined in *Shah et al., 2015*, and purified with a Zymo RNA concentrator kit. gRNAs were diluted to 1 µg/µl and stored at –80°C. Both guides were mixed with Cas9 protein (PNA Bio, Newbury Park, CA) and simultaneously injected into embryos at the single-cell stage. Isolation and sequencing of a single allele showed a 15 base pair insertion containing a premature stop codon 234 base pairs into the second exon (sequence with insert underlined, stop codon in bold: GACCTTCTGT **TGA**TCGGACCTTCGGGTGGCCCGTCTC), resulting in a truncated protein of 88 amino acids. Genotyping of mutant animals was conducting using a standard PCR protocol with primers (listed 5′ to 3′): fwd: GCTGCCCGGCATTACACATCTC; rev: GCTCAGCGGTTGGAGGTGGATA (Integrated DNA Technologies).

Tg[*myo6b:mitoGFP*][w213] fish express GFP targeted to the mitochondrial matrix via a cytochrome C oxidase subunit VIII localization sequence driven by an HC-specific promoter. The mitoGFP construct was created using the Gateway Tol2 system (Invitrogen) under the HC-promoter *myosin6b*, injected into single-cell embryos, and maintained as genetic lines.

## Serial block-face scanning electron microscopy

Fish were fixed in 4% glutaraldehyde prepared with 0.1 M sodium cacodylate (pH 7.4) overnight, then decapitated. Heads were fixed for an additional night. Samples were then processed using a modified version of the Ellisman protocol (*J. Deerinck et al., 2022*). Briefly, samples were treated in 4% osmium tetroxide for 1 hr. After washout, samples were infiltrated with epon resin and baked at 60°C for 48 hr. Transverse semithin sections were taken beginning at the anterior of the fish to localize NMs (*Raible and Kruse, 2000*). The beginning of an NM was identifiable by the presence of a small clustering

of SCs. Samples were then mounted onto a pin and placed into the SBFSEM Volumescope (Apreo, Thermo Fisher Scientific). Images were collected with a pixel size of 5 nm. Slices were 40–50 nm thick. HCs, SCs, and their mitochondria were reconstructed in TrakEM2.0 (*Cardona et al., 2012*) via manual segmentation. HC ribbons were defined as structures with a dark center surrounded by vesicles. Volume measurements were performed in AMIRA 6.5 for EM Systems (Thermo Fisher Scientific). Measures of kinocilium and stereocilium length were performed in TrakEM2.0. Where appropriate, indications of young or old HCs were based off the linear regression of kinocilium to stereocilia length. Measurements of object position within the NM were performed in TrakEM 2.0, and their relative geometric distances calculated using a Microsoft Excel script. No masking was used during analysis. If not all HCs from an NM could be segmented, HCs were chosen randomly such that they filled out the developmental age range via stereocilia length.

## Waterjet and calcium imaging

A waterjet assay (*Pickett et al., 2018*) was used to stimulate HCs and record their calcium responses. Imaging was conducted at ambient temperature (25–26°C). Fish (5–6 dpf) were immobilized with 0.2% MESAB and positioned ventral-side up in EM solution under a harp so that NMs OC1, D1, or D2 (*Raible and Kruse, 2000*) were accessible. NMs were imaged using an inverted Marianas spinning disk system (Intelligent Imaging Innovations, 3i) and a Zeiss C-Apochromat 63×/1.2 NA water immersion objective. NMs were first located under brightfield. A glass pipette filled with EM was placed approximately 100 μm from the NM. Proper positioning of the waterjet pipette was confirmed by visualizing movement of the kinocilia at the apical end of the NM. After a 6 s baseline, a 10 Hz sinusoidal pressure wave was applied using a pressure clamp (HSPC-1, ALA Scientific) for 20 s. Images were taken alternating between a 488 nm laser (exposure time 100 ms) and a 561 nm laser (exposure time 250 ms) in 1 s intervals using Slidebook (Intelligent Imaging Innovations). Timelapses were analyzed in Slidebook and Microsoft Excel. Cells that demonstrated signal rundown during baseline were omitted from analysis.

## Confocal microscopy

TMRE dye (Invitrogen) was prepared according to manufacturer's specifications. Tg[*myo6b:mitoG-FP*]$^{w213}$ fish (5 dpf) were allowed to swim freely in EM containing the dye (1 nM) for 1 hr prior to imaging, then immobilized with 0.2% MESAB and mounted onto coverslips with 1.5% ultrapure agarose. Primary posterior lateral line NMs were imaged with an Zeiss 880 confocal with Airyscan technology and a 40× water immersion objective. Z-stacks were taken through the NMs in 0.22 μm steps. Whole NMs were analyzed using IMARIS software (Oxford Instruments) by creating 3D masks from the green channel to measure mean fluorescence from the red channel. *opa1* mutants were identified based on the readily distinguishable mitochondrial fragmentation observed in high-resolution fluorescence microscopy.

## Data analysis

Statistical analyses were conducted using GraphPad Prism 9.5.0 software (Dotmatics). For all comparisons, we used parametric tests only if the data fit a normal distribution (D'Agostino and Pearson test) and the variances were not statistically different (F-test). Otherwise, we used nonparametric tests. For comparison of multiple groups, we used a Kruskal–Wallis test with Dunn's multiple comparisons. For comparisons of most SBF datasets, we used Kolmogorov–Smirnov tests to best distinguish differences between whole distributions. For live imaging, we used Mann–Whitney tests to best distinguish differences in the distribution medians.

PCA and two-dimensional UMAP analysis were conducted and statistically analyzed using Python 3.9.7. PCA was performed using the Python package scikit-learn (*Pedregosa et al., 2012*). UMAP was performed with the package umap-learn (*McInnes et al., 2018*) and exploratory spatial analytics with the packages ESDA and PySAL (*Rey and Anselin, 2007*). For PCA, we used the following variables: (1) number of mitochondria, (2) the total mitochondrial volume, (3) volume of the max mito, (4) volume of the median mitochondrion, (5) z-score of max mito, (6) max mito cable length, (7) average minimum distance of largest mitochondrion to the ribbons, and (8) average minimum distance of the median mitochondrion to the ribbons. To determine whether stereocilia length was distributed nonrandomly across the manifold, we calculated Moran's I. To determine whether genotypes differentially

distributed, we performed join count analysis. For all statistical tests, an alpha value of 0.05 was considered statistically significant.

## Acknowledgements

The authors thank David White and the UW Fish Facility staff for animal care. We thank Teresa Nicolson for the *cdh23* and *cav1.3a* mutants. The Tg[*myo6b:mitoGFP*]^w213 line was created by Sarah Pickett. Lastly, the authors thank Rachel Wong for initially providing equipment and training in SBF reconstructions, and for thoughtful comments on the manuscript.

## Additional information

### Funding

| Funder | Grant reference number | Author |
|---|---|---|
| National Institute on Deafness and Other Communication Disorders | RO1DC015783 | David W Raible |
| National Institute on Deafness and Other Communication Disorders | F32DC017343 | Andrea McQuate |

The funders had no role in study design, data collection and interpretation, or the decision to submit the work for publication.

### Author contributions

Andrea McQuate, Conceptualization, Data curation, Formal analysis, Funding acquisition, Investigation, Methodology, Writing – original draft, Project administration; Sharmon Knecht, Resources; David W Raible, Conceptualization, Resources, Data curation, Formal analysis, Supervision, Funding acquisition, Investigation, Methodology, Project administration, Writing - review and editing

### Author ORCIDs

Andrea McQuate ⬥ http://orcid.org/0000-0002-2052-2004
David W Raible ⬥ http://orcid.org/0000-0002-5342-5841

### Ethics

All experiments were done in compliance with the University of Washington Institutional Animal Use and Care Committee (IACUC protocol number 2997-01).

### Decision letter and Author response

Decision letter https://doi.org/10.7554/eLife.80468.sa1
Author response https://doi.org/10.7554/eLife.80468.sa2

## Additional files

### Supplementary files
• MDAR checklist

### Data availability

The SBFSEM data volumes for the sixteen different NMs used in this manuscript, including WT and mutants, are publicly available at webKnossos (http://demo.webknossos.org/), searchable by the dataset name indicated in the relevant source data.

The following datasets were generated:

| Author(s) | Year | Dataset title | Dataset URL | Database and Identifier |
|---|---|---|---|---|
| McQuate AM, Knecht S, Raible D | 2023 | 03052021_WT_left_2 | https://webknossos.org/datasets/91969da369e1a848/03052021_WT_left_2/view#4743,4995,263,0,1.3 | Webknossos, 4743,4995,263,0,1.3 |
| McQuate AM, Knecht S, Raible D | 2023 | 03052021_WT_SO1_Right | https://webknossos.org/datasets/91969da369e1a848/03052021_WT_SO1_Right/view#4925,5724,246,0,1.3 | Webknossos, 4925,5724,246,0,1.3 |
| McQuate AM, Knecht S, Raible D | 2023 | 03052021_WT_SO2_Right | https://webknossos.org/datasets/91969da369e1a848/03052021_WT_SO2_Right/view#4774,5185,341,0,1.3 | Webknossos, 4774,5185,341,0,1.3 |
| McQuate AM, Knecht S, Raible D | 2023 | 02102020_WT_6dpf_Right_3 | https://webknossos.org/datasets/91969da369e1a848/02102020_WT_6dpf_Right_3/view#4930,5395,267,0,1.3 | Webknossos, 4930,5395,267,0,1.3 |
| McQuate AM, Knecht S, Raible D | 2023 | 02102020_WT_6dpf | https://webknossos.org/datasets/91969da369e1a848/02102020_WT_6dpf/view#4947,5529,222,0,1.3 | Webknossos, 4947,5529,222,0,1.3 |
| McQuate AM, Knecht S, Raible D | 2023 | 02102020_WT_3dpf_C | https://webknossos.org/datasets/91969da369e1a848/02102020_WT_3dpf_C/view#4953,3852,212,0,1.3 | Webknossos, 4953,3852,212,0,1.3 |
| McQuate AM, Knecht S, Raible D | 2023 | 02102020_WT_3dpfA_2 | https://webknossos.org/datasets/91969da369e1a848/02102020_WT_3dpfA_2/view#5297,5328,323,0,1.3 | Webknossos, 5297,5328,323,0,1.3 |
| McQuate AM, Knecht S, Raible D | 2023 | 11262018_Opa1 | https://webknossos.org/datasets/91969da369e1a848/11262018_Opa1/view#3332,3547,205,0,1.3 | Webknossos, 3332,3547,205,0,1.3 |
| McQuate AM, Knecht S, Raible D | 2023 | 12052019_cdh23_right | https://webknossos.org/datasets/91969da369e1a848/12052019_cdh23_right/view#4623,4852,195,0,1.3 | Webknossos, 4623,4852,195,0,1.3 |
| McQuate AM, Knecht S, Raible D | 2023 | 12052019_cdh23_left | https://webknossos.org/datasets/91969da369e1a848/12052019_cdh23_left/view#5129,5047,160,0,1.3 | Webknossos, 5129,5047,160,0,1.3 |

*Continued on next page*

*Continued*

| Author(s) | Year | Dataset title | Dataset URL | Database and Identifier |
|---|---|---|---|---|
| McQuate AM, Knecht S, Raible D | 2023 | 12192019_cdh23_IO3 | https://webknossos.org/datasets/91969da369e1a848/12192019_cdh23_IO3/view#5690,5783,282,0,1.3 | Webknossos, 5690,5783,282,0,1.3 |
| McQuate AM, Knecht S, Raible D | 2023 | 12192019_cdh23_fA_SO1 | https://webknossos.org/datasets/91969da369e1a848/12192019_cdh23_fA_SO1/view#5232,6593,288,0,1.3 | Webknossos, 5232,6593,288,0,1.3 |
| McQuate AM, Knecht S, Raible D | 2023 | 07292020_CaV_A_Right | https://webknossos.org/datasets/91969da369e1a848/07292020_CaV_A_Right/view#5932,6878,248,0,1.3 | Webknossos, 5932,6878,248,0,1.3 |
| McQuate AM, Knecht S, Raible D | 2023 | 07292020_CaV_A_Left | https://webknossos.org/datasets/91969da369e1a848/07292020_CaV_A_Left/view#6142,7029,322,0,1.3 | Webknossos, 6142,7029,322,0,1.3 |
| McQuate AM, Knecht S, Raible D | 2023 | 07292020_CaV_fishB_right | https://webknossos.org/datasets/91969da369e1a848/07292020_CaV_fishB_right/view#4836,5799,367,0,1.3 | Webknossos, 4836,5799,367,0,1.3 |
| McQuate AM, Knecht S, Raible D | 2023 | 07302020_cav_fishB_left_1 | https://webknossos.org/datasets/91969da369e1a848/07302020_cav_fishB_left_1/view#5657,5364,352,0,1.3 | Webknossos, webknossos.org |

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
