## [Editor Report]

This valuable study of serial block-face scanning electron microscopy on zebrafish lateral line hair cells provided compelling cellular evidence for the importance of normal hair cell function in establishing mitochondrial patterning. This work will be of broad interest to cell biologists studying mitochondrial function.

---

## [Decision Letter]

**Decision letter after peer review:**

Thank you for submitting your article "Activity regulates a cell type-specific mitochondrial phenotype in zebrafish lateral line hair cells" for consideration by *eLife*. Your article has been reviewed by 3 peer reviewers, and the evaluation has been overseen by a Reviewing Editor and Didier Stainier as the Senior Editor. The following individual involved in the review of your submission has agreed to reveal their identity: Katie Kindt (Reviewer #1).

Essential revisions:

All three reviewers thought this high-resolution mitochondrial study in sensory hair cells is novel and interesting. Nevertheless, there are some valid concerns raised by the reviewers that should be improved such as:

1) A better link between mitochondrial disruption to hair cell function in opa1 mutants,

2) Improve data analyses, and

3) Better functional evidence to support the hypothesis based on morphological analysis.

We will be happy to look at a revision of this manuscript should you be able to address the reviewers' concerns.

*Reviewer #1 (Recommendations for the authors):*

This is a rich dataset and resource for the community that has been thoroughly and thoughtfully analyzed. The presentation of the data and figures is high quality and thoughtfully executed. Our comments seek to further strengthen the manuscript.

1. Please provide a table(s) to summarize the EM sample data, including the age, neuromast identity/identities, genotype, whether the samples are whole or partial neuromasts, which cells have been reconstructed (HCs, SCs), and a list of the figures that use data from that sample.

2. Include information in your introduction that highlights the total number of samples reconstructed and the public availability of the data. This is an amazing resource for the community and is worth highlighting more.

3. For cdh23 and cav1.3 mutants, please provide a comparison of wt to mutant mitochondrial volume/total cell volume as in Figure S1B. It looks like the HC cell size may be different in some of the mutants, which may impact mitochondrial volume.

4. Sample sizes. How many different fish were analyzed for P-SCs? The number of NMs and fish are different in different figure legends. If n = 1 fish, there should be more than 1 NM and more than 1 fish analyzed for P-SC mitos. It would seem that the data already exists (as there are 2+ fish for both HCs and C-SCs), so it should be possible to at least increase these numbers to 2 NMs and 2 fish for all cell types. More broadly, multiple fish would ideally be examined for each mutant. We recognize that this is incredibly labor-intensive work, but recent EM datasets published in e*Life* have shown data for n>1 even in mutants (ex. Dow et al. 2018).

5. Can you show how the position of the max mito changes over developmental time for Figures 3 and 4?

6. In a supplement it would be helpful to add a plot showing the sizes of all mitochondria for a mature, representative HC(s)? It would be nice to see an example(s) of the full distribution.

7. Opa1 mutations have a broad range of effects, as mitochondrial fusion is hypothesized to impact many aspects of mitochondrial function. This makes it hard to link functional changes in these mutants to changes in mito architecture. Please discuss this caveat.

8. Although evoked mitoGCaMP3 responses in opa-1 mutants are compelling, changes in evoked responses could also be due to defects in mechanotransduction. Another method to verify normal mechanotransduction would be helpful. For example, normal FM 1-43 labeling or unaltered evoked cyto- or mem-GCaMP6 responses.

*Reviewer #2 (Recommendations for the authors):*

Results and Figures:

– Labeling of larval age and hair cell age and position is inconsistent throughout the results and figures. For example, in Figure 2 A neither larval age nor hair cell position are specified.

– Line 139-140: "Younger hair cells on periphery and mature hair cells in center" need a citation.

– Development of mitochondria with maturation (Figure 3): the authors show images of representative neuromasts from younger (3 dpf) and more mature (5-6 dpf) neuromasts in Figure 3A, yet analyzed pooled mitochondrial morphologies from all three ages using the position of the hair cell in the neuromast as a proxy for maturity. Given that 3 dpf neuromasts are not yet functionally mature and do not show concentric patterning comparable to older neuromasts, it is unclear why the authors used this approach. The analysis shown in Figure 3B-E would be more convincing if central and peripheral hair cell data were shown for just 5-6 dpf, with 3 dpf shown separately.

– Figure 4A: How were young, intermediate, and mature hair cells defined?

– Figure 5D-F: Is this data pooled from all 3 ages?

– UMAP data/ Figure 6 and Figure 10: A more detailed description of UMAP data analysis and what it specifically reveals about mitochondrial properties in relation to hair cell maturation and/or function is needed.

– opa1 mutants/ Figure 7: The authors show reduced evoked mitochondrial calcium uptake in opa1 mutants. Did they examine mitoGCaMP levels at rest (i.e. compare mitoGCaMP fluorescence in WT vs mutant without stimulation)? Additionally, there is a possibility that the hair cells in opa1 mutants have impaired mechanotransduction. Did the authors examine FM1-43 uptake in mutants vs. WT?

– cdh23 mutants/ Figure 8: To support the conclusion that the largest mitochondria do not localize to the presynaptic ribbons, the authors should show the localization of the ribbons in the representative reconstructions in Figure 8A.

– cav1.3a mutants/ Figure 9: This analysis overall would be far more impactful if it corresponded with the analysis performed on cdh23 mutants. The authors should repeat the analysis of the data in the same way it was performed for cdh23 mutants, then expand on the cav1.3a mutant analysis with an additional figure.

– UMAP data/ Figure 10: In Figure 10A, does the WT condition contain cells from all ages (3 and 5/6 dpf)? If so, that additional variable may be confounding, as the two mutants examined only represent hair cells from 5 dpf fish.

Discussion:

– The discussion on the possible mechanisms for distinct mitochondrial morphology at the apical and basolateral ends of hair cells needs some revising. It's unclear to this reviewer how the data shown supports that calcium signaling shapes mitochondrial morphology through fission. The authors propose that calcium influx through MET may itself contribute to smaller mitochondria, yet large amounts of calcium enter presynaptic Cav1.3a channels when active hair cells are stimulated, and this is where the mitochondrial volume is the largest. Conversely, while the authors suggest that large mitochondrial networks near the synapses may be specifically necessary to drive PMCA1 pumps, it should be noted that PMCA2 pumps play a critical role in the regulation of calcium at hair cell stereocilia. Overall, it seems that other key differences, such as the high metabolic demand of synaptic transmission itself, may play a role in the localization of large mitochondria to synapses.

Materials and methods:

– More details are needed for waterjet and calcium imaging. What concentration of MESAB was used? What was the composition of the extracellular solution? Which neuromast was acquired?

*Reviewer #3 (Recommendations for the authors):*

1. Some data from the too small sample size. In particular, P-SC in Figure 1 and 2, and Opa1 mutant in Figure 6A, and B are from only 1 fish. Although the authors pointed out the laborious nature of the reconstructions and suggested that their data reflect the greater population in the Discussion section, the reviewer believes that their reconstruction method and results are the central topics of this study. Therefore, the description of too much difficulty and speculation will make their promising data and techniques unattractive. In addition, the authors have a concern about the possibility of mixed genotype (WT and het) in lines 445 – 447. Even though previous another project has reported no impact on results, is this speculation scientifically accepted? There is no guarantee whether heterozygous was used or as WT or not.

2. The representative images will help to understand the quantified data in Figures 3B – E (mitochondria), Figure 5 (Ribbon), and Figure 7F (data from Opa1 mutant). In Figure 3, the reviewer could not consider whether the data from 3 – 6 dpf can be combined without the images. In addition to the image, please consider the statistical method in Figures 5A – C, although the conclusion won't be affected. The authors analyzed the data by one-way ANOVA with Tukey's test, however, two-way ANOVA will be applicable in this case because central and peripheral cells are paired. Figure 7F and F' seem to be identical. Please confirm the images.

3. The overall data, the authors analyzed using the cells from 3, 5, 6 dpf. Some data was pooled from all developmental stages and others were separated. It should be consistent throughout the study. The authors explained that 3 dpf neuromasts did not yet have a concentric patterning (lines 144 – 145). If so, how do the authors distinguish central and peripheral in Figure 5H? If they can distinguish, the data of 3 dpf in Figure 5H can be split into peripheral and central to fit the same style with 5 and 6dpf data. While 5 and 6 dpf data are treated differently depending on the data. The data from 5 and 6 dpf are mostly analyzed separately but not in Figure 5H. Moreover, the regression curve to show the correlation between kinocilium length and mitochondria or ribbon was only 5 dpf. If the author wants to discuss the development, the data 3, 5, and 6 dpf need to be analyzed separately and all data should be in the manuscript as main or supplemental figures.

4. The description of Figures 8F and G in the text (lines 245 – 246) does not match to the figure. The text says two-way ANOVA but the figure legend says Kruskal-Wallis test. Adding WT data to Figure 8E (even if it is the same data as Figure 2) and performing Two-way ANOVA will help the reader to understand.

5. In the Discussion section, the authors are trying to interpret their morphological data by comparing previous functional analyses, however, the discussion includes many speculations about the mitochondrial function. In addition to TMRE and Ca measurement in Opa1 mutant, the author needs to address some of the functions using their models if they want to focus on the mitochondrial functions. For instance ROS measurement, ATP production, biogenesis markers analysis, or response to neomycin as described in the introduction section comparing each developmental stage (3, 5, 6 dpf) or among cell types (hair cell, central, peripheral…).

Esterberg et al., J Clin Invest. 2016 (https://www.jci.org/articles/view/84939)

Alassaf et al., *eLife*, 2019 (https://elifesciences.org/articles/47061#s2)

Hirose et al., Hearing Research, 2016 (https://www.sciencedirect.com/science/article/pii/S0378595516301599?via%3Dihub)

[Editors’ note: further revisions were suggested prior to acceptance, as described below.]

Thank you for resubmitting your work entitled "Activity regulates a cell type-specific mitochondrial phenotype in zebrafish lateral line hair cells" for further consideration by *eLife*. Your revised article has been evaluated by Didier Stainier (Senior Editor) and a Reviewing Editor.

The manuscript has been much improved. While the reviewers agreed that no additional experiments are needed, specific suggestions for clarification are outlined below.

*Reviewer #1 (Recommendations for the authors):*

I have completed my review of the revised manuscript by McQuate et al.

The authors have done an admirable job addressing the suggestions provided by all the reviewers. On its own, this work is a well-rounded, complete body of work. In addition, this work represents an immense resource for the field. I do not have any major concerns moving forward-the paper is suitable for publication.

*Reviewer #2 (Recommendations for the authors):*

The consistency of the analysis and data presentation was much improved in this revised manuscript. Yet a detailed interpretation of what the UMAP analysis reveals about mitochondrial properties in relation to hair cell maturation and in the context of the two mutants was not included in this revision. It is important for the authors to unpack for the readers what they think their UMAP analysis reveals and how it supports their hypothesis.

- The authors did not adequately interpret their UMAP analysis in Figure 5 nor explain what they think it reveals about mitochondrial properties in relation to hair cell maturation. Stating that older hair cells have "a significantly different mitochondrial phenotype than younger hair cells" is too vague. For example, there appear to be differences in the clustering of hair cell mitochondrial morphologies based on stereocilia length (Figure 5 A) and chronological age of the NM (Figure 5 B), yet there is an area of clear overlap in clustering on the left side where all three chronological ages overlap (Figure 5 B) in an area with shorter stereocilia length (Figure 5 A). This reviewer thinks that detail may provide the support that functional maturation at all three chronological ages plays a key role in establishing hair cell mitochondrial morphology, but if that's the case, it should be discussed by the authors in the results.

- Similarly, a more detailed interpretation is needed of what the UMAP analysis in Figure 9 reveals about the differences in cdh23-/- vs. cav1.3a-/- hair cell mitochondria.

- The terms "young" and "mature" (5 dpf) or "older" (3 dpf) to describe hair cells are used throughout the manuscript, yet the authors state there are no categorical values for maturity. Some measurable criteria (i.e. range of stereocilia lengths) must have been used for the authors to identify cells as "young" vs "older/mature", and these stereocilia length parameters should be defined in the methods.

- Better cohesion on the discussion of how calcium may be influencing the size of mitochondria is needed. In line 333 the authors identify calcium influx at the apical end as potentially driving the smaller size of mitochondria, yet don't address that the largest mitochondria are also found near synaptic ribbons with high levels of calcium influx until line 402. Speculation on how calcium may be influencing these two cellular compartments differently should be included in the discussion, but they should be in one section and discussed in relation to one another.

*Reviewer #3 (Recommendations for the authors):*

The revised manuscript has been improved a lot and is well aligned.

Data and figures are revised to make the readers understand easily.

The reviewer found one critical mislabel in Figures 6B and C.

WT and Opa1 data seem to be switched. So, the reviewer would like the author to confirm it.

Finally, even though the author analyzed many mitochondria, the author did not increase the number of opa1 mutant fish (used only 1 fish for analysis) but emphasized the data robustness. However, the reviewer is still wondering whether the data from n = 1 can indicate robustness, support the scientific meaning, and convince the reader.

The *eLife* transparent reporting form asks authors to state information about sample sizes (which should include details of the methods used to estimate them and the assumptions made).

https://elifesciences.org/articles/21070 *eLife* journal confirms and admits, it will be no big problem.

---

## [Author Response]

Essential revisions:All three reviewers thought this high-resolution mitochondrial study in sensory hair cells is novel and interesting. Nevertheless, there are some valid concerns raised by the reviewers that should be improved such as:1) A better link between mitochondrial disruption to hair cell function in opa1 mutants,

We have completed new experiments using a cytoplasmic calcium indicator in addition to the mitochondrial calcium indicator for waterjet experiments comparing WT and opa1 mutants. We find that there is no significant difference between WT and mutants in cytoplasmic calcium changes in response to stimulation, suggesting that mechanotransduction is not substantially altered. By contrast, mitochondrial calcium increases after stimulation is significantly reduced in mutants compared to WT.

2) Improve data analyses, and

We have made major changes to improve data analysis and better support our overall conclusions. (1) We removed data from partially sectioned neuromasts and added additional data from newly reconstructed complete neuromasts so that all analyses could be consistent throughout. (2) We analyzed additional neuromasts from more fish to ensure our data is representative. (3) We re-analyzed mitochondrial changes related to hair cell maturation by including new measurements of stereocilia length, previously shown by Kindt et al. 2012 to be a robust marker of hair cell age analogous to kinocilium length. We include correlative analysis of stereocilia length and kinocilium length supporting this previous study, and now consistently apply this metric across our current study. We therefore no longer use categorical variables such as stage or position as a proxy for age. (4) We re-organized figures to improve consistency throughout the paper so that it is easier to compare wildtype and mutant animals. While we believe that our analysis is much more robust, our overall conclusions are unchanged from the previous submission.

3) Better functional evidence to support the hypothesis based on morphological analysis.

We certainly recognize that there are many additional functional studies suggested by our current morphological analysis. We believe that the functional data we do include has been improved (see comment 1 above). However, our current study is by no means insubstantial and indeed is unprecedented in its detail. We believe it will serve as an important resource for additional studies.

We will be happy to look at a revision of this manuscript should you be able to address the reviewers' concerns.Reviewer #1 (Recommendations for the authors):This is a rich dataset and resource for the community that has been thoroughly and thoughtfully analyzed. The presentation of the data and figures is high quality and thoughtfully executed. Our comments seek to further strengthen the manuscript.1. Please provide a table(s) to summarize the EM sample data, including the age, neuromast identity/identities, genotype, whether the samples are whole or partial neuromasts, which cells have been reconstructed (HCs, SCs), and a list of the figures that use data from that sample.

As noted above we now include data only from whole neuromasts, and summarize this data collection in Table 1. We have completely reorganized figures to make clear which datasets are used in which figures. In addition, we provide the names of each dataset and how they precisely relate to each figure in the supplementary material. We further note that these datasets have been made publicly available. We also include all source data.

2. Include information in your introduction that highlights the total number of samples reconstructed and the public availability of the data. This is an amazing resource for the community and is worth highlighting more.

We thank the Reviewer for appreciating the sheer magnitude of work that went into this study, and its usefulness for the scientific community. Overall, we reconstructed 5,908 mitochondria from 162 cells from 16 neuromasts from 10 fish. We added this summary in a Table, along with a statement of public availability in the introduction (lines 90-99).

3. For cdh23 and cav1.3 mutants, please provide a comparison of wt to mutant mitochondrial volume/total cell volume as in Figure S1B. It looks like the HC cell size may be different in some of the mutants, which may impact mitochondrial volume.

We now include cell volumes for *cdh23* mutants (Figure 7C) and *cav1.3* mutants (Figure 8C). While HCs are on average slightly larger in mutants than in WT, there is no statistical difference in the ratio of mitochondrial volume to cell volume in these different genotypes.

4. Sample sizes. How many different fish were analyzed for P-SCs? The number of NMs and fish are different in different figure legends. If n = 1 fish, there should be more than 1 NM and more than 1 fish analyzed for P-SC mitos. It would seem that the data already exists (as there are 2+ fish for both HCs and C-SCs), so it should be possible to at least increase these numbers to 2 NMs and 2 fish for all cell types. More broadly, multiple fish would ideally be examined for each mutant. We recognize that this is incredibly labor-intensive work, but recent EM datasets published in eLife have shown data for n>1 even in mutants (ex. Dow et al. 2018).

We now include data for both peripheral and central SCs from 3 neuromasts in 2 different fish and note this in the text (line 112).

We now include data for *cdh23* and *cav1.3a* mutants from 4 different neuromasts in 2 different fish, and note this in the text-

*cdh23*: line 252.

*cav1.3a:* line 270.

It is an honor to have our work compared to the landmark Dow et al. 2018 paper. This study was primarily concerned with the HC body, and thus only contains reconstructions of whole HCs. Our manuscript differentiates from the previous study by individually reconstructing each mitochondrion within HCs (on average around 40 per HC), often in addition to the cell body. The manual reconstruction and individual quantification of each mitochondrion is where the bulk of the laborious nature of our study lies.

5. Can you show how the position of the max mito changes over developmental time for Figures 3 and 4?

We now show changes in the position of the max mito during development for 5-6 dpf WT (Figure 2H), 3 dpf WT (Figure 4H), *cdh23* (Figure 7H), and *cav1.3a* (Figure 8H).

6. In a supplement it would be helpful to add a plot showing the sizes of all mitochondria for a mature, representative HC(s)? It would be nice to see an example(s) of the full distribution.

We’ve included this in Figure 2—figure supplement 2.

7. Opa1 mutations have a broad range of effects, as mitochondrial fusion is hypothesized to impact many aspects of mitochondrial function. This makes it hard to link functional changes in these mutants to changes in mito architecture. Please discuss this caveat.

We thank the reviewer for pointing out this important caveat when interrupting mitochondrial fusion, and have added it to the discussion (Lines 342-343).

8. Although evoked mitoGCaMP3 responses in opa-1 mutants are compelling, changes in evoked responses could also be due to defects in mechanotransduction. Another method to verify normal mechanotransduction would be helpful. For example, normal FM 1-43 labeling or unaltered evoked cyto- or mem-GCaMP6 responses.

We thank the reviewer for suggesting this important control. We have now repeated waterjet experiments measuring both cytoplasmic and mitochondrial calcium responses in wildtype and mutant animals (Figure 6). We find no significant difference in cytoplasmic calcium responses between mutant and WT animals, but see a marked reduction in mitochondrial calcium responses in mutants compared to WT.

Reviewer #2 (Recommendations for the authors):Results and Figures:– Labeling of larval age and hair cell age and position is inconsistent throughout the results and figures. For example, in Figure 2 A neither larval age nor hair cell position are specified.

We thank the Reviewer for pointing this out. As summarized above, we have completely redesigned this analysis of the relationships between HC maturity and mitochondrial and ribbon phenotypes, and we believe it is now more consistent and clear.

– Line 139-140: "Younger hair cells on periphery and mature hair cells in center" need a citation.

We have removed the analysis of position and maturity.

– Development of mitochondria with maturation (Figure 3): the authors show images of representative neuromasts from younger (3 dpf) and more mature (5-6 dpf) neuromasts in Figure 3A, yet analyzed pooled mitochondrial morphologies from all three ages using the position of the hair cell in the neuromast as a proxy for maturity. Given that 3 dpf neuromasts are not yet functionally mature and do not show concentric patterning comparable to older neuromasts, it is unclear why the authors used this approach. The analysis shown in Figure 3B-E would be more convincing if central and peripheral hair cell data were shown for just 5-6 dpf, with 3 dpf shown separately.

We thank the Reviewer for noting the confusing presentation of the data in the previous submission. The 5-6dpf data are consistently pooled throughout the manuscript, with the 3 dpf data having its own separate figure. The only places all three ages are pooled are in the UMAP analysis, where we are specifically analyzing the HC mito developmental trajectory.

– Figure 4A: How were young, intermediate, and mature hair cells defined?

We no longer assign categorical values for maturity.

– Figure 5D-F: Is this data pooled from all 3 ages?

This analysis has changed.

– UMAP data/ Figure 6 and Figure 10: A more detailed description of UMAP data analysis and what it specifically reveals about mitochondrial properties in relation to hair cell maturation and/or function is needed.

We have added more description of the analysis to the manuscript text (lines 201-204) and additional details of how the test was conducted to the Methods (lines 509-518).

– opa1 mutants/ Figure 7: The authors show reduced evoked mitochondrial calcium uptake in opa1 mutants. Did they examine mitoGCaMP levels at rest (i.e. compare mitoGCaMP fluorescence in WT vs mutant without stimulation)?

We thank the Reviewer for pointing out baseline calcium mitochondrial levels could be different between WT and mutant. We now include a comparison baseline fluorescence in Figure 6 —figure supplement 1, and show there was no significant difference between WT and mutant.

Additionally, there is a possibility that the hair cells in opa1 mutants have impaired mechanotransduction. Did the authors examine FM1-43 uptake in mutants vs. WT?

As noted in the response to Reviewer 1, we have now analyzed both cytoplasmic and mitochondrial calcium responses in mutants, and show that mutants have no significant impairment of mechanotransduction.

– cdh23 mutants/ Figure 8: To support the conclusion that the largest mitochondria do not localize to the presynaptic ribbons, the authors should show the localization of the ribbons in the representative reconstructions in Figure 8A.

We thank the Reviewer for pointing out how to make this conclusion stronger. Images for all genotypes now include ribbons.

– cav1.3a mutants/ Figure 9: This analysis overall would be far more impactful if it corresponded with the analysis performed on cdh23 mutants. The authors should repeat the analysis of the data in the same way it was performed for cdh23 mutants, then expand on the cav1.3a mutant analysis with an additional figure.

We have completely revised the analysis to be more comprehensive. Moreover we have made figure layouts of wildtype and mutant analysis more consistent.

– UMAP data/ Figure 10: In Figure 10A, does the WT condition contain cells from all ages (3 and 5/6 dpf)? If so, that additional variable may be confounding, as the two mutants examined only represent hair cells from 5 dpf fish.

Removal of the 3 dpf data does not impact the results of the UMAP (now included as Figure 9 —figure supplement 2). We’ve included the 3 dpf data in the main figure so that it sheds light on the developmental trajectory.

Discussion:– The discussion on the possible mechanisms for distinct mitochondrial morphology at the apical and basolateral ends of hair cells needs some revising. It's unclear to this reviewer how the data shown supports that calcium signaling shapes mitochondrial morphology through fission. The authors propose that calcium influx through MET may itself contribute to smaller mitochondria, yet large amounts of calcium enter presynaptic Cav1.3a channels when active hair cells are stimulated, and this is where the mitochondrial volume is the largest. Conversely, while the authors suggest that large mitochondrial networks near the synapses may be specifically necessary to drive PMCA1 pumps, it should be noted that PMCA2 pumps play a critical role in the regulation of calcium at hair cell stereocilia. Overall, it seems that other key differences, such as the high metabolic demand of synaptic transmission itself, may play a role in the localization of large mitochondria to synapses.

We thank the Reviewer for pointing out this interesting paradox! We have softened these speculations to indicate calcium pumps may be only one of several aspects of synaptic transmission needing metabolic support, and further emphasized the metabolic demand of synaptic transmission possibly driving max mito formation (lines 345-349).

Materials and methods:– More details are needed for waterjet and calcium imaging. What concentration of MESAB was used? What was the composition of the extracellular solution? Which neuromast was acquired?

Added (Lines 480-491). The composition of the external solution (EM) is listed in line 439-440.

Reviewer #3 (Recommendations for the authors):1. Some data from the too small sample size. In particular, P-SC in Figure 1 and 2,

We now include additional supporting cell data from more cells across several fish.

and Opa1 mutant in Figure 6A, and B are from only 1 fish.

We now emphasize that the *opa1* phenotype is robust and visible through fluorescence, as seen in the supplemental figure. We believe the EM reconstruction is sufficient to make our point.

Although the authors pointed out the laborious nature of the reconstructions and suggested that their data reflect the greater population in the Discussion section, the reviewer believes that their reconstruction method and results are the central topics of this study. Therefore, the description of too much difficulty and speculation will make their promising data and techniques unattractive.

We have re-edited the discussion to soften this point.

In addition, the authors have a concern about the possibility of mixed genotype (WT and het) in lines 445 – 447. Even though previous another project has reported no impact on results, is this speculation scientifically accepted? There is no guarantee whether heterozygous was used or as WT or not.

We have removed these data and replaced it with reconstruction of a new neuromast that is WT. For the Reviewer’s interest, we note that there was no change in the conclusion.

2. The representative images will help to understand the quantified data in Figures 3B – E (mitochondria), Figure 5 (Ribbon), and Figure 7F (data from Opa1 mutant). In Figure 3, the reviewer could not consider whether the data from 3 – 6 dpf can be combined without the images.

We thank the Reviewer for pointing out how to strengthen the presentation of these data. We have completely re-done figures and believe that the images now clearly represent the quantified data.

In addition to the image, please consider the statistical method in Figures 5A – C, although the conclusion won't be affected. The authors analyzed the data by one-way ANOVA with Tukey's test, however, two-way ANOVA will be applicable in this case because central and peripheral cells are paired.

We have changed this analysis and no longer use categorical labels.

Figure 7F and F' seem to be identical. Please confirm the images.

We note that these images were NOT identical. However, we have now replaced them with other images to make this point clearer to the casual reader.

3. The overall data, the authors analyzed using the cells from 3, 5, 6 dpf. Some data was pooled from all developmental stages and others were separated. It should be consistent throughout the study. The authors explained that 3 dpf neuromasts did not yet have a concentric patterning (lines 144 – 145). If so, how do the authors distinguish central and peripheral in Figure 5H? If they can distinguish, the data of 3 dpf in Figure 5H can be split into peripheral and central to fit the same style with 5 and 6dpf data. While 5 and 6 dpf data are treated differently depending on the data. The data from 5 and 6 dpf are mostly analyzed separately but not in Figure 5H. Moreover, the regression curve to show the correlation between kinocilium length and mitochondria or ribbon was only 5 dpf. If the author wants to discuss the development, the data 3, 5, and 6 dpf need to be analyzed separately and all data should be in the manuscript as main or supplemental figures.

We recognize that the previous presentation was confusing and therefore made extensive changes to the analysis. Foremost is the use of stereocilia length to measure hair cell maturity, which vacated the requirement for categorical analysis. We removed data from incomplete neuromasts and added new data to expand the analysis. We now directly compare data from different age fish as well to strengthen our conclusions. All data is included in the manuscript, and in addition we include all source data. We note that none of our conclusions have changed from the previous submission.

4. The description of Figures 8F and G in the text (lines 245 – 246) does not match to the figure. The text says two-way ANOVA but the figure legend says Kruskal-Wallis test. Adding WT data to Figure 8E (even if it is the same data as Figure 2) and performing Two-way ANOVA will help the reader to understand.

We thank the Reviewer for pointing out that this was confusing. Given the non-normality of these data, we opted to change this analysis to only comparing differences in max-mito percentage in the basal-most HC quadrant using a non-parametric test, which we deemed more appropriate than a parametric two-way ANOVA. We included the new comparison in all relevant figures.

5. In the Discussion section, the authors are trying to interpret their morphological data by comparing previous functional analyses, however, the discussion includes many speculations about the mitochondrial function.

We believe that interpreting our data in light of previous published studies is not excessively speculative. We believe we have been both balanced and accurate. Indeed, comparing reported results to other studies is the fundamental underlying function of a Discussion section of any paper.

In addition to TMRE and Ca measurement in Opa1 mutant, the author needs to address some of the functions using their models if they want to focus on the mitochondrial functions. For instance ROS measurement, ATP production, biogenesis markers analysis, or response to neomycin as described in the introduction section comparing each developmental stage (3, 5, 6 dpf) or among cell types (hair cell, central, peripheral…).Esterberg et al., J Clin Invest. 2016 (https://www.jci.org/articles/view/84939)Alassaf et al., eLife, 2019 (https://elifesciences.org/articles/47061#s2)Hirose et al., Hearing Research, 2016 (https://www.sciencedirect.com/science/article/pii/S0378595516301599?via%3Dihub)

While we agree with the Reviewer that these additional studies would be interesting, they are beyond the scope of the current paper. We believe that the manuscript as it stands includes substantial data and analysis that we certainly hope will spark future study.

[Editors’ note: further revisions were suggested prior to acceptance, as described below.]

Reviewer #2 (Recommendations for the authors):The consistency of the analysis and data presentation was much improved in this revised manuscript. Yet a detailed interpretation of what the UMAP analysis reveals about mitochondrial properties in relation to hair cell maturation and in the context of the two mutants was not included in this revision. It is important for the authors to unpack for the readers what they think their UMAP analysis reveals and how it supports their hypothesis.- The authors did not adequately interpret their UMAP analysis in Figure 5 nor explain what they think it reveals about mitochondrial properties in relation to hair cell maturation. Stating that older hair cells have "a significantly different mitochondrial phenotype than younger hair cells" is too vague.- Similarly, a more detailed interpretation is needed of what the UMAP analysis in Figure 9 reveals about the differences in cdh23-/- vs. cav1.3a-/- hair cell mitochondria.

We thank the Reviewer for the comment, and have added additional text to clarify why we performed PCA analysis and accompanying UMAP dimensionality reduction. We realize that separating the PCA analyses into two separate figures was potentially confusing to readers, so instead we have combined Figures 5 and 9 into one figure.

Our overall goal was to test whether we saw a distinction between wildtype and mutant HCs with respect to mitochondrial properties when we examined all these properties in aggregate. However these measurements likely have co-variance, so we used PCA, as principal components are by definition uncorrelated. We then represented the multidimensional PCA space as a projection in two dimensions using UMAP. Finally we performed spatial autocorrelation analysis to test whether stereocilia length was differentially distributed across the UMAP projection and then used neighbor analysis to test whether genotypes differentially clustered.

We have combined figures 5 and 9 into one figure (new Figure 8). Additional text describing the rationale and analysis ins found in lines 275-286.

For example, there appear to be differences in the clustering of hair cell mitochondrial morphologies based on stereocilia length (Figure 5 A) and chronological age of the NM (Figure 5 B), yet there is an area of clear overlap in clustering on the left side where all three chronological ages overlap (Figure 5 B) in an area with shorter stereocilia length (Figure 5 A). This reviewer thinks that detail may provide the support that functional maturation at all three chronological ages plays a key role in establishing hair cell mitochondrial morphology, but if that's the case, it should be discussed by the authors in the results.

We provided evidence that there are hair cells with varying degrees of maturity at different developmental stages when comparing 3dpf to 5-6dpf hair cells in Figure 4. As suggested, we have added additional text here to again highlight this point.

- The terms "young" and "mature" (5 dpf) or "older" (3 dpf) to describe hair cells are used throughout the manuscript, yet the authors state there are no categorical values for maturity. Some measurable criteria (i.e. range of stereocilia lengths) must have been used for the authors to identify cells as "young" vs "older/mature", and these stereocilia length parameters should be defined in the methods.

This information can be found in Figure 2- Supplement Figure 1, Figure 4-Supplement Figure 1, and Figure 8-Supplement Figure 1. Stereocilia length was used as a continuous measure of HC age throughout the manuscript, and the labels directly reflect the linear regression shown in the aforementioned Figures, which is why the stereocilia length for HCs in the Main Figures is included alongside the labels “young” or “mature.” We have added a line to the methods to clarify this point.

- Better cohesion on the discussion of how calcium may be influencing the size of mitochondria is needed. In line 333 the authors identify calcium influx at the apical end as potentially driving the smaller size of mitochondria, yet don't address that the largest mitochondria are also found near synaptic ribbons with high levels of calcium influx until line 402. Speculation on how calcium may be influencing these two cellular compartments differently should be included in the discussion, but they should be in one section and discussed in relation to one another.

The two points are addressed in sequential paragraphs, and the conundrum is pointed out in line 408:

“This suggests a paradox where calcium might promote both mitochondrial fission apically and fusion basally. However, synaptic transmission requires large energy expenditures in addition to calcium regulation, and these demands may instead drive mitochondrial fusion into basal networks. Understanding the paradoxical effects of HC activity influencing both the formation of smaller apical mitochondria and larger basal networks will require additional study.”

Reviewer #3 (Recommendations for the authors):The revised manuscript has been improved a lot and is well aligned.Data and figures are revised to make the readers understand easily.The reviewer found one critical mislabel in Figures 6B and C.WT and Opa1 data seem to be switched. So, the reviewer would like the author to confirm it.

We thank the reviewer for pointing out this mistake. We’ve corrected it.

Finally, even though the author analyzed many mitochondria, the author did not increase the number of opa1 mutant fish (used only 1 fish for analysis) but emphasized the data robustness. However, the reviewer is still wondering whether the data from n = 1 can indicate robustness, support the scientific meaning, and convince the reader.The eLife transparent reporting form asks authors to state information about sample sizes (which should include details of the methods used to estimate them and the assumptions made).https://elifesciences.org/articles/21070 eLife journal confirms and admits, it will be no big problem.

We clarify (lines 201- 208) that mutations in *opa1* cause mitochondrial fragmentation from yeast to humans. We state that the zebrafish phenotype is readily observable and completely penetrant by fluorescent imaging. The n = 1 fish included 5 hair cells, and is in regards to the SBF reconstruction, which is only included in this figure to make the point definitive. Physiology experiments used multiple neuromasts from multiple fish.